# UniCon: Unidirectional Information Flow for Effective Control of Large-Scale Diffusion Models

**Fanghua Yu[1], Jinjin Gu[2], Jinfan Hu[1], Zheyuan Li[1], Chao Dong[1,3]**[*]

[1]Shenzhen Institutes of Advanced Technology, Chinese Academy of Sciences
[2]The University of Sydney    [3]Shanghai AI Laboratory
`fanghuayu96@gmail.com,{jf.hu1,zy.li3,chao.dong}@siat.ac.cn,`
`jinjin.gu@sydney.edu.au`

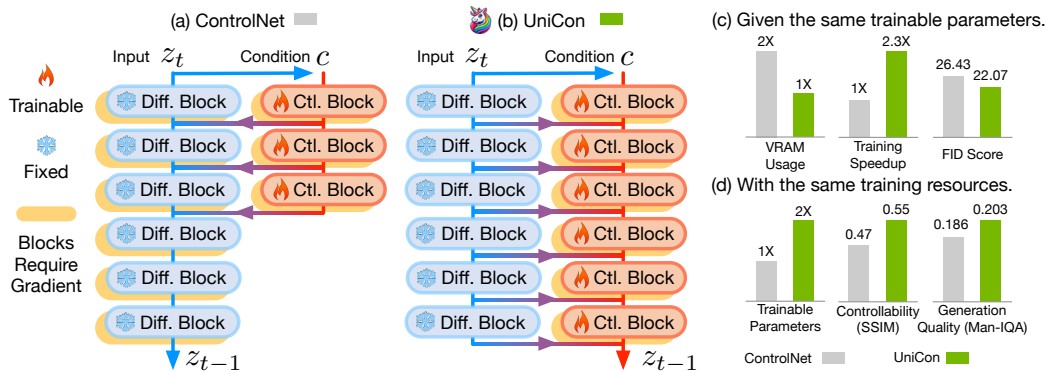

Figure 1: This figure illustrates the schematic comparison between our proposed UniCon and ControlNet. In UniCon, information flows unidirectionally from the diffusion model to the adapter network, which directly outputs the results. This design is highly computationally efficient as it does not require computing and storing gradients for the diffusion model. (c) displays results generated from downsampled images, and (d) shows outcomes based on depth maps. UniCon achieves improved performance while utilizing fewer resources.

## Abstract

We introduce UniCon, a novel architecture designed to enhance control and efficiency in training adapters for large-scale diffusion models. Unlike existing methods that rely on bidirectional interaction between the diffusion model and control adapter, UniCon implements a unidirectional flow from the diffusion network to the adapter, allowing the adapter alone to generate the final output. UniCon reduces computational demands by eliminating the need for the diffusion model to compute and store gradients during adapter training. Our results indicate that UniCon reduces GPU memory usage by one-third and increases training speed by 2.3 times, while maintaining the same adapter parameter size. Additionally, without requiring extra computational resources, UniCon enables the training of adapters with double the parameter volume of existing ControlNets. In a series of image conditional generation tasks, UniCon has demonstrated precise responsiveness to control inputs and exceptional generation capabilities.

## 1 Introduction

Diffusion generative models (Ho et al., 2020; Song et al., 2020; Nichol & Dhariwal, 2021; Rombach et al., 2022), with their exceptional generative effects and diversity, have significantly impacted computer vision fields, such as creative design (Anciukevičius et al., 2023; Mittal et al., 2021; Cao et al., 2024), image processing (Wang et al., 2023b; Lin et al., 2023; Yang et al., 2023; Yu et al., 2024), and personalized content generation (Zhang et al., 2023; Mou et al., 2024; Li et al., 2024;

---

[*]Corresponding author (e-mail: chao.dong@siat.ac.cn)

2023; Zhao et al., 2024). This is largely attributed to their ability to precisely control complex layouts, poses, shapes, and image conditions using both textual and visual prompts. These functions typically require training an additional adapter or controller network associated with the diffusion model, enabling control of the output during the inference process (Zhang et al., 2023; Mou et al., 2024; Zavadski et al., 2023; Zhao et al., 2024). The most successful representative is ControlNet (Zhang et al., 2023) for the Stable Diffusion (SD) model (Rombach et al., 2022), which includes a trainable copy of the SD U-Net encoder part and zero convolution layers as connectors. To achieve higher quality generation and more precise control, the parameter size of diffusion models has also been significantly increased (Podell et al., 2023; Lin et al., 2024; Sauer et al., 2023). Starting from the original SD model (Rombach et al., 2022) with 0.86 billion parameters, the advanced SD3 (Esser et al., 2024) has scaled up to 8 billion parameters. The latest models (Peebles & Xie, 2023; Bao et al., 2023; Esser et al., 2024) replaced the commonly used U-Net architecture with transformer-based models. The benefits of scaling up have not yet reached their limit. We may expect larger diffusion models and advanced transformer architectures to appear in the future (Xie et al., 2023; Han et al., 2023; Luo et al., 2023; Chen et al., 2024a). Under this trend, the limitations of existing control adapters for diffusion models have become increasingly apparent.

There are three main problems: First, as the parameters of the diffusion model increase, the size of its adapter also needs to be expanded accordingly. Existing methods not only calculate the gradients of the adapters during training, but also calculate and store the gradients of the diffusion model, placing significant strain on computational resources. This typically implies nearly double the additional overhead. Second, while existing adapter designs exert control by modifying features in intermediate layers, they still process these modified features using the original parameters of the diffusion model. The generative capabilities of models with fixed parameters are limited when only the intermediate features are modified. Third, the design of existing control adapters assumes that the diffusion models use an encoder-decoder architecture (*i.e.* U-Net), which is inadequate for transformer-based diffusion models due to their inability to separate encoder and decoder components. Since existing adapters primarily implement control within the encoder part, their response to control signals lacks pixel-level precision, particularly problematic in high-precision tasks such as image restoration based on low-quality images (Yu et al., 2024).

In this paper, we introduce a novel design specifically tailored for training control adapters for the next generation of large-scale diffusion models. This design facilitates further scaling up of diffusion models and their control. Unlike the existing methods (Zhang et al., 2023; Zavadski et al., 2023) that use adapters to intervene during the forward inference of diffusion models in a bidirectional manner, our approach allows information to flow unidirectionally from the diffusion network to the adapter, without flowing backward. The final output image will be generated by the adapter, NOT the diffusion model. Our method is therefore called UniCon. Figure 1 illustrates the design difference between UniCon and previous methods. The advantage of UniCon is that during training, the diffusion model only needs to perform forward propagation and does not need to compute or store gradients. Additionally, UniCon can utilize all parameters and architectures of the diffusion model, not just those pertaining to the encoder, making it suitable for diffusion models with large-scale transformer architecture. Moreover, since the output processing is handled by the adapter, our design enables more precise control over generation based on conditions. UniCon avoids micro-interventions in the diffusion model architecture, thus it could further enhance the versatility.

We test the UniCon architecture across a variety of conditional generative tasks, validating it on both the SD U-Net diffusion model and the DiT diffusion model. Notably, when applying the UniCon adapter to the transformer-based DiT diffusion model, our approach saves half of the video memory (VRAM) usage while achieving a 2.3X increase in training speed, without increasing the adapter's parameter size. It also delivers superior FID scores and condition fidelity in tasks involving image generation from downsampled images, see Figure 1 (c). By employing our method in the generation task according to Canny edge, the parameter size of the adapter can be doubled without additional computational resources. UniCon can further improve performance, where our method not only enhances the quality of generation but also ensures optimal controllability.

## 2 RELATED WORK

**Diffusion generative models.** The diffusion models (Sohl-Dickstein et al., 2015) have been explored across a variety of generative tasks, such as image-to-image translation (Zhao et al., 2022),

text-to-image synthesis (Rombach et al., 2022; Saharia et al., 2022a; Avrahami et al., 2022; Jiang et al., 2022), image restoration (Saharia et al., 2022b; Daniels et al., 2021; Kawar et al., 2022; Lin et al., 2023; Yu et al., 2024; Tao et al., 2024), image editing (Meng et al., 2021; Avrahami et al., 2022; 2023), image inpainting (Nichol et al., 2021; Lugmayr et al., 2022), etc. Model scaling up has been an important means to improve the capabilities of these models, and a lot of efforts have been made, *e.g.* a series of works proposed by Stability AI, including SD-2.1 (Rombach et al., 2022), SD-XL (Podell et al., 2023), SD-Cascade (Pernias et al., 2023), and SD3 (Esser et al., 2024). In order to further expand and improve the capacity of the model, the large-scale transformer architecture is also introduced for diffusion models, including DiT (Peebles & Xie, 2023), U-ViT (Bao et al., 2023), SD3 (Esser et al., 2024) and the PixArt family (Chen et al., 2023; 2024b;a).

**Controlled generation of diffusion models.** Pre-trained large-scale diffusion models serve as foundational models that can be fine-tuned for a variety of downstream tasks. The fine-tuning process should be designed to mitigate issues such as overfitting, model collapse, and catastrophic forgetting. Various fine-tuning methods have been developed across multiple fields, including HyperNetworks (Alaluf et al., 2022; Dinh et al., 2022), additive learning (Rosenfeld & Tsotsos, 2018; Mallya et al., 2018; Mallya & Lazebnik, 2018; Serra et al., 2018), and Low-Rank Adaptation (LoRA) (Hu et al., 2021). For diffusion models, the most effective and commonly used approach for enhancing generation quality and controllability involves training an additional adapter network (Houlsby et al., 2019; Stickland & Murray, 2019; Li et al., 2022; Mou et al., 2024; Ju et al., 2024; Mo et al., 2023). Zhang *et al.* (Zhang et al., 2023) first introduced ControlNet, an architecture designed to incorporate spatial conditioning controls into large, pre-trained diffusion models. The ControlNet is connected with zero-initialized convolution layers that progressively grow the parameters from zero and ensure that no harmful noise could affect the fine-tuning. Building on this, ControlNet-XS (Denis Zavadski & Rother, 2023) explored different sizes and architectural designs of ControlNet to enhance control over the image generation process in stable diffusion-based models. Additionally, T2I-Adapter (Mou et al., 2024) focused on aligning internal knowledge within T2I models with external control signals, while keeping the original large T2I models unchanged. This approach allows for the training of various adapters under different conditions, such as text and image, to achieve detailed control and editing capabilities in the color and structure of the generated images. Uni-ControlNet (Zhao et al., 2024) integrates various local and global controls into a single model, using only two additional adapters for fine-tuning on pre-trained diffusion models, thus avoiding the extensive costs of training from scratch. Furthermore, surpassing existing diffusion-based restoration methods (Wang et al., 2023b; Lin et al., 2023; Yang et al., 2023), SUPIR (Yu et al., 2024) leverages the generative prior and the potential of large diffusion model SDXL (Podell et al., 2023) to enable photo-realistic restoration of severely degraded images through textual prompts. This illustrates the significant potential of diffusion models in enhancing image quality and fidelity in controlled generation tasks.

## 3 METHOD

**Preliminary and motivation.** A typical diffusion model involves two processes: a forward process that gradually adds a small amount of noise to the image, and a corresponding backward denoising process that recovers the input image by gradually removing the noise. Given a pre-trained diffusion model $\mathcal{H}(\cdot)$, it can either have a U-Net-like structure, as in SD, or a transformer structure, as in the DiT model. When performing the backward generation process, the diffusion model takes the last denoised result $z_t$ and time step $t$ as well as the text prompt $p$ as input, and predicts the next result of denoising generation $z_{t-1} = \mathcal{H}(z_t, t, p)$. The existing adapter methods directly modify the features $X_h$ of the intermediate layer of $\mathcal{H}$ during the denoising generation process to modify the generation results according to conditions. Assume we have an adapter $\mathcal{C}(\cdot)$, usually it may take $z_t$, $t$, and $p$ as inputs. Additionally, it may also take $X_h$ as inputs (Zavadski et al., 2023). The output of the adapter $\mathcal{C}$ is a series of residual modified values $X_c$ of the intermediate layers $X_h$. In ControlNet, $X_c$ will be directly added to the corresponding intermediate layer of $\mathcal{H}$ using element-wise addition to generate a new representation $X_h' = X_c + X_h$ to implement conditional generation.

This intuitive paradigm aligns with the common practice of fine-tuning large pre-trained diffusion models, yet it introduces several significant issues:

- Firstly, this approach directly impacts the diffusion model; therefore, training the adapter involves calculating the gradients of the modified diffusion model and subsequently backpropagating these gradients to the adapter's trainable parameters. To achieve rapid and stable convergence, the

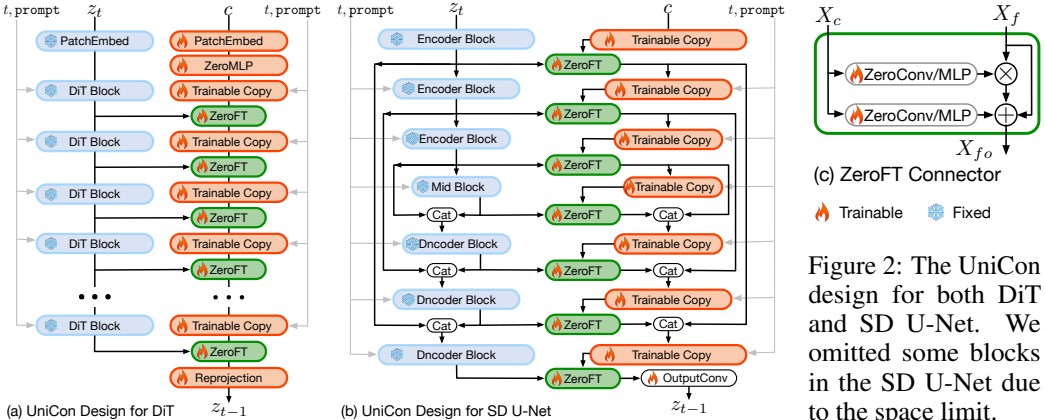

Figure 2: The UniCon design for both DiT and SD U-Net. We omitted some blocks in the SD U-Net due to the space limit.

adapter's initial parameters are set as a trainable copy of the diffusion model. This can lead to training costs for the adapter surpassing those of the diffusion model itself. For example, when training ControlNet for DiT, when the gradient of ControlNet itself occupies about 18GB VRAM, the gradient brought by the DiT diffusion model needs to occupy 16GB VRAM, and this cost can be optimized. As the parameter number and resource demands of diffusion models continue to grow, training large adapters for large-scale models poses significant engineering challenges.

- Secondly, while modifying the features of intermediate layers can directly alter the output, this technique still depends on pre-trained parameters to process these changes. When parameters are fixed, adapting the model to new data by only changing the input of each layer restricts its generative capabilities. Alternative methods such as LoRA (Hu et al., 2021) or full-parameter fine-tuning permit direct modifications of parameters but also come with limitations. LoRA is constrained by its scale, and full-parameter tuning risks losing previously learned generative capabilities.

- Finally, the existing design of control adapters, which often assumes a U-Net-like encoder-decoder diffusion architecture (Zhang et al., 2023; Mou et al., 2024; Zavadski et al., 2023; Zhao et al., 2024) and focuses primarily on the encoder (Zhang et al., 2023), does not suit for transformer-based diffusion models. This mismatch arises because it is difficult to distinctly separate the encoder and decoder components in such models. Indiscriminately controlling the entire transformer model escalates resource demands due to the necessity of computing and storing the diffusion model's gradients. Furthermore, since existing adapters mainly focus on the encoder, their response to control signals lacks pixel-level precision, particularly problematic in high-precision tasks such as generation based on low-quality images.

**Unidirectional information flow design paradigm.** We propose a design paradigm named UniCon to address these issues. For a clearer understanding, readers are encouraged to refer to Figure 1. Our approach ensures that the pre-trained diffusion model $\mathcal{H}$ is used only for inference, thereby eliminating the need to compute and store its gradients. Information flows *unidirectionally* from $\mathcal{H}$ to the UniCon adapter $\mathcal{C}_{UC}$, positioning the adapter as a trainable decoder that processes information from the diffusion model's layers. Specifically, denote $X_h = x_{h1}, x_{h2}, \ldots$ as the output from the intermediate layers of $\mathcal{H}$. The UniCon adapter receives these intermediate features along with the inputs of the condition $c$, time step $t$, and prompt $p$. The output of the adapter under this step is the denoising result $z_{t-1} = \mathcal{C}_{UC}(c, t, p, X_h)$. This design establishes a unidirectional information flow path from $z_t \rightarrow \mathcal{H}(\cdot) \rightarrow X_h \rightarrow \mathcal{C}_{UC}(\cdot) \rightarrow z_{t-1}$, ensuring that information does not revert to the pre-trained diffusion model. By mirroring the architecture and parameters of the diffusion model in the adapter and using a zero-initialized connector for integration, we can ensure stable training.

This design resolves issues found in existing approaches. Firstly, the diffusion model only participates in forward inference, eliminating the need for computing and storing its gradients, which significantly reduces computational costs. Secondly, all adapter parameters are trainable, and the adapter directly delivers processed outputs, bypassing the fixed parameters of the diffusion model. At the same time, the well-trained diffusion model remains unchanged, ensuring the generative capabilities are less likely to be forgotten during fine-tuning. This enhances the generative capabilities and leads to improved generation quality. Moreover, as there is no need to intervene in the diffusion model – only to extract the computed features from intermediate layers – our solution is highly adaptable to

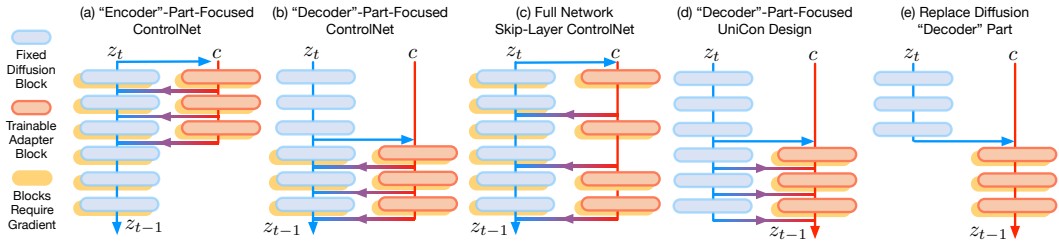

Figure 3: Schematic representation of the five different variants we covered in our ablation studies.

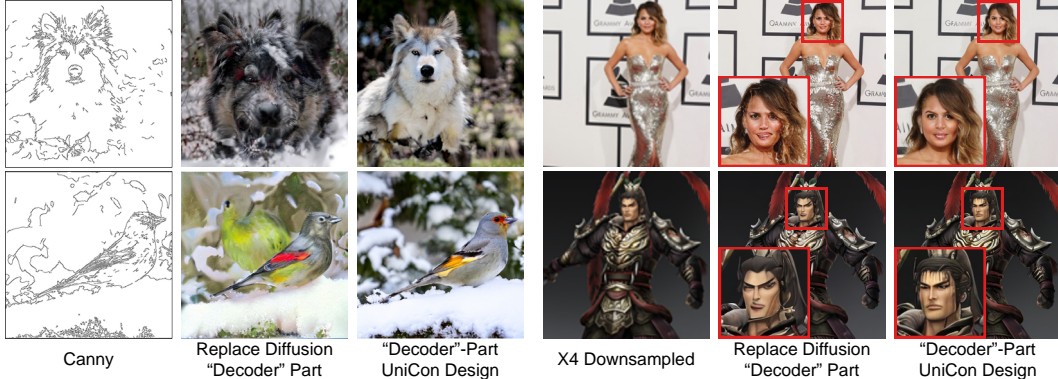

Figure 4: The comparison between decoder-part-focused UniCon and replace diffusion decoder part. The results indicate that if the complete pre-trained diffusion model is not preserved, there is a significant decline in generative capabilities. These two models are shown in Figure 3 (d) and (e).

different diffusion architectures, from U-Net to transformer model with ease. Finally, the adapter no longer relies solely on the encoder or parts of the diffusion model but instead produces the output directly, further improving the preservation of control signals in tasks requiring high precision.

**The proposed UniCon adapter designs.** In this work, we use DiT and SD U-Net to validate our UniCon adapter paradigm. Nevertheless, we note that the UniCon design paradigm is widely applicable and can be employed across various diffusion models. Figure2(a) and Figure2(b) show the structural details when UniCon is applied to DiT and SD U-Net. The design of the UniCon adapter, applicable to both DiT and U-Net models, starts with a trainable copy that includes all parameters and architectures of the diffusion model. In order to connect the diffusion model to the adapter, a connector module is also required. For each output $x_h$ of a DiT block or U-Net block, it introduces information from $x_h$ to the corresponding location in the adapter. Zero-initialized convolutional layers (ZeroConv) or fully connected layers (ZeroMLP) are simple and stable choices for U-Net and DiT. This approach prevents unwanted noise from affecting features in trainable replicas at the beginning of training. We additionally propose here a new connector called Zero-initialization Feature Transform (ZeroFT). ZeroFT not only utilizes element-wise addition but also incorporates element-wise multiplication and a shortcut connection, enhancing the integration and transfer of information. This setup is detailed in Figure2(c). The parameters in both two layers within ZeroFT – whether convolutional or fully-connected layers – are initialized to zero. According to our experiments, ZeroFT is more effective than ZeroMLP/Conv in generating effects and controlling performance.

## 4 EXPERIMENTS

### 4.1 EXPERIMENT SETTINGS

**Tasks and datasets.** We employ five different conditions for image generation for testing: canny edge (Canny, 1986), depth maps (Ranftl et al., 2021), OpenPose (Cao et al., 2019), 4× down-sampling (SR) (Wang et al., 2018), and blurring (sigma=2) followed by 4x downsampling (de-blur+downsampling) (Kong et al., 2022; Yu et al., 2024). These conditions can broadly be classified into two categories: (1) tasks that require the model to generate high-level semantic content, such as OpenPose, depth maps, and canny edge, which we refer to as high-level control generation tasks;

| Task | ControlNet Size | Controllability↑ | FID↓ | Clip-Score↑ |
|---|---|---|---|---|
| Canny (SSIM) | Encoder | 0.4748 | 51.52 | 0.7724 |
| | Decoder | **0.5131** | 59.32 | 0.7507 |
| | Skip-Layer | 0.4983 | **49.78** | 0.7776 |
| | Full | 0.5053 | 50.17 | **0.7818** |
| SR (PSNR) | Encoder | 34.82 | 26.43 | 0.7996 |
| | Decoder | 34.85 | 25.84 | 0.8013 |
| | Skip-Layer | 35.49 | 24.99 | 0.8009 |
| | Full | **36.53** | **23.04** | **0.8026** |

(a) The results of copy different parts in ControlNet.

| Task | Connector Type | Controllability↑ | FID↓ | Clip-Score↑ |
|---|---|---|---|---|
| Canny (SSIM) | ZeroMLP | 0.5343 | 55.22 | 0.7612 |
| | ShareAttn | 0.5236 | 56.22 | 0.7606 |
| | ZeroFT | **0.5426** | **52.31** | **0.7696** |
| SR (PSNR) | ZeroMLP | **35.67** | 22.99 | 0.8013 |
| | ShareAttn | 35.55 | 23.03 | 0.8012 |
| | ZeroFT | 35.64 | **22.07** | **0.8025** |

(b) The results of different connector designs used in the UniCon Adapter.

| Task | Adapter | Unidirectional | Controllability↑ | Generation Quality | | | | Text Consistency |
|---|---|---|---|---|---|---|---|---|
| | | | | FID↓ | Clip-IQA↑ | MAN-IQA↑ | MUSIQ↑ | Clip-Score↑ |
| Canny (SSIM) | Skip-Layer | ✗ | 0.4983 | **49.78** | **0.6629** | **0.1978** | **66.05** | **0.7776** |
| | | ✓ | **0.5078** | 56.93 | 0.6224 | 0.1737 | 63.66 | 0.7561 |
| | Decoder | ✗ | 0.5131 | 59.32 | 0.6047 | 0.1621 | 62.51 | 0.7507 |
| | | ✓ | **0.5343** | **55.22** | **0.6347** | **0.1780** | **64.27** | **0.7612** |
| | Full | ✗ | 0.5053 | 50.17 | 0.6397 | 0.1867 | 64.70 | 0.7818 |
| | | ✓ | **0.5458** | **46.71** | **0.6577** | **0.2029** | **66.45** | **0.7889** |
| SR (PSNR) | Decoder | ✗ | 34.85 | 25.84 | 0.6979 | 0.2325 | 68.26 | 0.8013 |
| | | ✓ | **35.59** | **23.55** | **0.7036** | **0.2358** | **68.61** | **0.8018** |
| | Full | ✗ | 36.53 | 23.04 | 0.7212 | 0.2609 | 69.91 | **0.8026** |
| | | ✓ | **37.34** | **20.34** | **0.7251** | **0.2831** | **69.99** | 0.8022 |

(c) The effect of the proposed unidirectional information flow design.

Table 1: Ablation study of different adapter designs. The diffusion model used in these experiments is DiT transformer-based model. ↑ indicates the larger the better and ↓ indicates the lower the better.

and (2) tasks that emphasize local generation and require higher precision in control, such as $4\times$ downsampling and blurring+downsampling, which we call low-level control generation. The images for training are selected from the LAION dataset (Schuhmann et al., 2022). We randomly select 2 million images with resolutions higher than $512 \times 512$. We center-crop and resize these images to $512 \times 512$, and pair them with the original text annotations from the LAION dataset to form image-text pairs. All images are pre-processed to compute the various conditions required for training, except for the Pose Condition. For the Pose Condition, we filter out images where no human pose was detected or key poses accounted for less than 30% of the body. We also exclude images with more than three people, as the pose detection error rate is higher in such cases.

**Implementation details.** In our experiments, we employed the PixelArt-$\alpha$ diffusion model as a representative example of the DiT model (Peebles & Xie, 2023) and StableDiffusion-2.1 (Rombach et al., 2022) as a representative example of the U-Net model. The training of the ControlNets was conducted using IDDPM (Nichol & Dhariwal, 2021), maintaining the same noise schedule as the diffusion models. All experiments were performed on four NVIDIA-RTX A6000 GPUs, employing the AdamW optimizer with a learning rate of $2 \times 10^{-4}$. The experiments were conducted with a total batch size of 64 and a total of 100,000 training steps.

**Evaluation.** We sample 1,000 images from the LAION Dataset for testing. Consistent with the training setting, the pose condition in the test set underwent the same image selection criteria. There are no duplicate images between the training and testing sets. To assess the controllability of different methods, we evaluate the alignment between the conditions of the generated images and the ground-truth conditions. Different metrics were chosen based on the type of condition: peak signal-to-noise ratio (PSNR) for low-level conditions; structural similarity (SSIM) (Wang et al., 2004) for canny edge conditions; mean square error (MSE) for depth conditions; and mean average precision (mAP) for object keypoint similarity in pose conditions (Zhao et al., 2024). In addition to assessing controllability, we evaluated the quality of generation using metrics such as FID (Heusel et al., 2017), Clip-IQA (Wang et al., 2023a), MAN-IQA (Yang et al., 2022), and MUSIQ (Ke et al., 2021). Furthermore, the Clip-Score (Hessel et al., 2021) was used to evaluate the consistency between generated images and their corresponding text prompts, ensuring an integrated and thorough assessment of both image quality and fidelity to conditioned inputs.

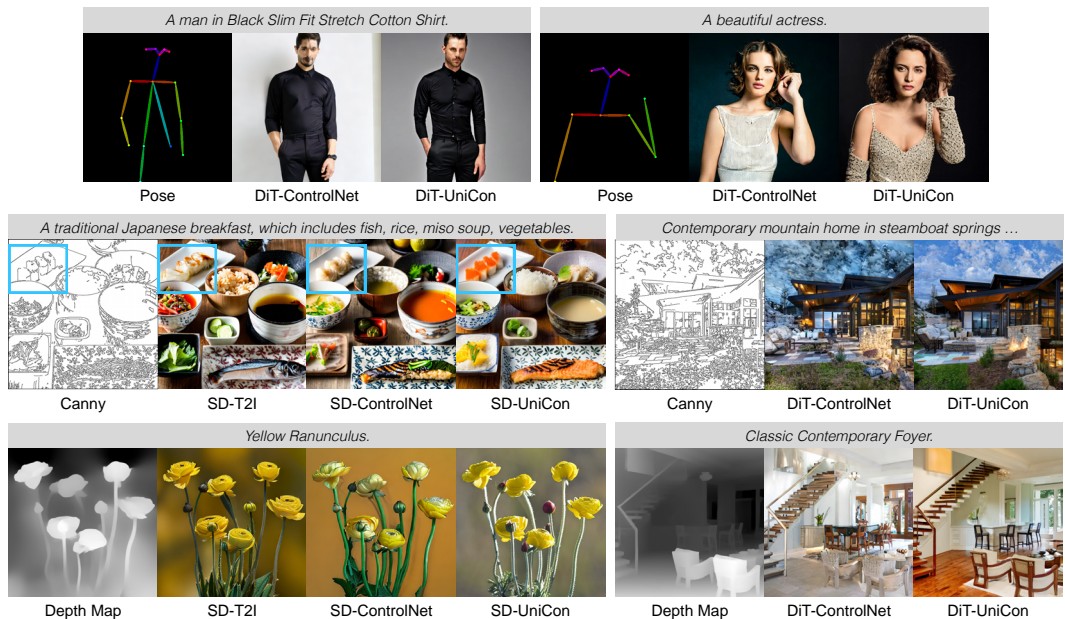

Figure 5: Comparison of different methods. We present the qualitative comparisons ControlNet (Zhang et al., 2023) and T2I (Mou et al., 2024) with both Stable Diffusion (SD) (Rombach et al., 2022) and Diffusion Transformer (DiT) (Peebles & Xie, 2023).

## 4.2 ABLATION STUDY

**The effect of copying "encoders" and copying "decoders" on diffusion control.** Using Control-Net as an example, we explored the influence of different parts of diffusion models on controlled generation. The design of ControlNet involves making a trainable copy of the network's "encoder". Based on the DiT architecture, we further designed three variants. The first variant involves making a trainable copy of and controlling the "decoder" part (the latter half) of the network, as shown in Figure 3(a). The second variant controls the entire network, covering both the encoder and decoder parts. To keep the parameter number constant, we also adopt the third variant of skipping and deleting every other block, as shown in Figure 3(c), referred to as the "Skip-Layer" design.

We test these variants on canny and SR tasks. The results are displayed in Table 1a. First, comparing different designs focusing on the encoder and decoder, the results indicate that while focusing on the encoder leads to better image quality, focusing on the decoder enhances controllability. This is probably because the understanding and generation of images and control signals are primarily conducted in the first half of the network, whereas the precise control of generation details occurs in the latter half. Specifically, for tasks such as SR that demand precise local control, a decoder-focused design might be more suitable. Given the distinct advantages of both encoder and decoder in controlling generation, we further explored the effects of utilizing both simultaneously. The results from the Skip-Layer variant confirmed this approach, demonstrating improved generative performance and controllability at the same parameter number. This proves that for DiT, distinguishing between the encoder and decoder is not effective, and we should leverage the capabilities of different parts of the entire diffusion model. The Full version, building upon the Skip-Layer, further increased the parameter number, and its enhanced generative performance and controllability also affirmed the importance of adapter capacity in controlling diffusion models.

**Unidirectional information flow *v.s.* changing feature directly.** The method proposed in this paper differs significantly from existing approaches by introducing the concept of unidirectional information flow. Maintaining the same adapter architecture and initialization, we opted to output using adapters rather than directly changing the features of the diffusion model. We test various adapter variants mentioned above, and the results are presented in Table 1c. Our results show that employing the unidirectional information flow for output substantially enhances performance,

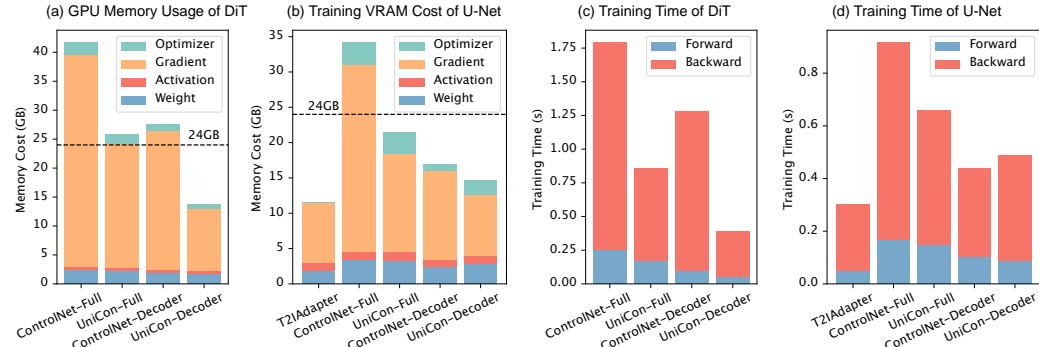

Figure 6: Comparison of training VRAM usage and training time of different adapter designs.

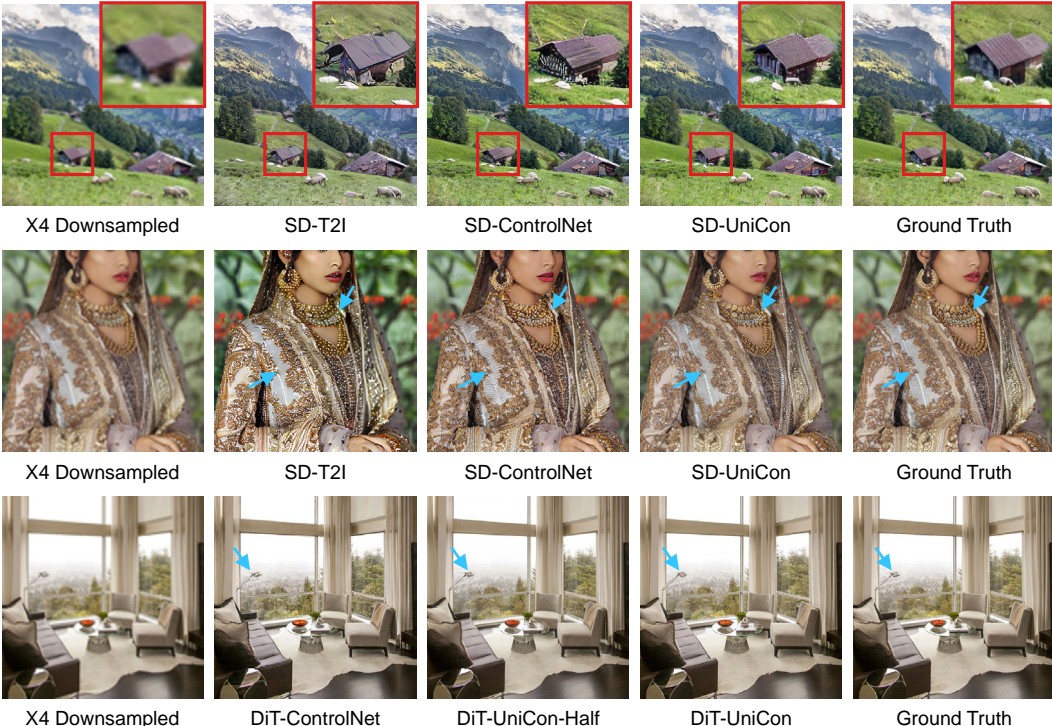

Figure 7: Comparison of different methods on conditional generation with low-level control inputs.

improving controllability and generative quality in both high-level and low-level tasks, whether using the diffusion model's "decoder" part or the full model as the adapter. For the "Skip-Layer" adapter, the unidirectional information flow design did not lead to performance gains. This is intuitively due to the skip-layer design compromising the output capability of the copied diffusion model. While the Skip-Layer design enhances the effectiveness of ControlNet, it is not suitable for UniCon. This ablation study demonstrates that a simple implementation of unidirectional information flow can significantly enhance both control effectiveness and generation quality.

**Training cost.** A major advantage of UniCon is its computational efficiency. As gradients do not need to be computed and stored for the diffusion model, UniCon significantly reduces VRAM usage and speeds up training. We compare VRAM occupation and training speed between UniCon and ControlNet in a single-GPU setup without any acceleration libraries. All experiments maintain consistency with standard training in terms of model parameters, diffusion settings. All necessary feature maps, such as text tokens, latent condition images, and latent ground-truth images, were pre-computed. We select the same 100 training batches, each with a size of 16, and all speed tests were performed on the same NVIDIA-RTX A6000 GPU. All parameters and computations were performed in BFloat16 precision. We detail the VRAM consumption as follows: (1) Weight: VRAM

| Diffusion Model | Task | Adapter | Controllability Metric | Value | FID↓ | Clip-IQA↑ | MAN-IQA↑ | MUSIQ↑ | Text Consistency Clip-Score↑ |
|---|---|---|---|---|---|---|---|---|---|
| DiT | Canny | ControlNet | SSIM↑ | 0.4748 | 51.52 | 0.6439 | 0.1861 | 65.24 | 0.7724 |
| | | UniCon | SSIM↑ | **0.5458** | **46.71** | **0.6577** | **0.2029** | **66.45** | **0.7889** |
| | Depth | ControlNet | MSE↓ | 84.65 | 53.63 | 0.6469 | 0.1838 | 64.88 | 0.7722 |
| | | UniCon | MSE↓ | **82.56** | **51.49** | **0.6514** | **0.2002** | **65.51** | **0.7785** |
| | Pose | ControlNet | mAP↑ | 0.4135 | 58.62 | 0.6332 | 0.1819 | 63.94 | 0.7430 |
| | | UniCon | mAP↑ | **0.4627** | **57.85** | **0.6600** | **0.1929** | **65.00** | **0.7545** |
| | SR | ControlNet | PSNR↑ | 34.82 | 26.43 | 0.7147 | 0.2459 | 69.37 | 0.7996 |
| | | UniCon-half | PSNR↑ | 35.64 | 22.07 | 0.7042 | 0.2675 | 69.51 | **0.8025** |
| | | UniCon | PSNR↑ | **37.34** | **20.34** | **0.7251** | **0.2831** | **69.99** | 0.8022 |
| | Deblur+ SR | ControlNet | PSNR↑ | 37.63 | 29.18 | **0.7199** | 0.2517 | 69.72 | 0.8004 |
| | | UniCon-half | PSNR↑ | 38.33 | 25.12 | 0.6998 | 0.2563 | 69.31 | **0.8014** |
| | | UniCon | PSNR↑ | **41.13** | **21.29** | 0.7089 | **0.2701** | **69.80** | 0.8012 |
| SD U-Net | Canny | ControlNet | SSIM↑ | 0.4895 | 49.80 | 0.6683 | 0.2215 | 67.19 | 0.8168 |
| | | T2I-Adapter | SSIM↑ | 0.3936 | 52.05 | **0.6783** | 0.2266 | **68.16** | 0.8155 |
| | | UniCon | SSIM↑ | **0.5570** | **47.11** | 0.6704 | **0.2335** | 67.99 | **0.8189** |
| | Depth | ControlNet | MSE↓ | 85.70 | 54.30 | 0.6828 | 0.2262 | 67.90 | 0.8202 |
| | | T2I-Adapter | MSE↓ | 87.72 | 55.09 | **0.6906** | **0.2331** | **68.12** | 0.8209 |
| | | UniCon | MSE↓ | **85.00** | **53.45** | 0.6807 | 0.2262 | 67.85 | **0.8214** |
| | SR | ControlNet | PSNR↑ | 31.66 | 30.19 | 0.7373 | 0.3266 | **70.94** | **0.8044** |
| | | T2I-Adapter | PSNR↑ | 18.94 | 48.20 | 0.6822 | 0.2795 | 70.91 | 0.7812 |
| | | UniCon-half | PSNR↑ | 34.38 | 28.29 | 0.7387 | 0.3244 | 69.76 | 0.8037 |
| | | UniCon | PSNR↑ | **35.69** | **22.80** | **0.7442** | **0.3271** | 70.48 | 0.8027 |

Table 2: Comparisons of different controllable diffusion models. ↑ indicates the larger the better and ↓ indicates the lower the better.

used just for loading the model. (2) Activation: VRAM used for network forward propagation without gradient computation. (3) Gradient: VRAM used to store gradients. (4) Optimizer: VRAM used for updating learnable parameters of the model. We also record the time costs for network forward propagation (FP) and backward propagation (BP). FP time accounted for the time spent on the diffusion model forward step, and BP time included the total time for updating gradients and optimizer parameters. Finally, we report the peak VRAM and average time costs over 100 iterations, with results presented in Figure 6. It can be seen that UniCon significantly reduces VRAM usage compared to ControlNet, saving nearly half the storage required for gradients, which are a major component of VRAM consumption during training. In terms of training time, because it eliminates the need to compute gradients for the diffusion model, the time spent on BP is also nearly halved.

**The choice of the connector.** We explored various methods for connecting diffusion models with adapters and found that different connectors lead to different outcomes. Based on the DiT diffusion model, we test three connector designs, including ZeroMLP, which is consistent with the ControlNet (Zhang et al., 2023; Li et al., 2024), ShareAttention (Esser et al., 2024; Zhang & Agrawala, 2024), which is suited for Transformer attention layers, and ZeroFT. The results of these designs are displayed in Table 1b. The results indicate that our proposed ZeroFT connector offers superior control and generative performance for both high-level and low-level tasks.

**Retaining the original diffusion model preserves the previously learned generative capabilities.** During the development of UniCon, a natural question emerged: Given that the outputs are already handled by the adapter, is it necessary to retain the pre-trained diffusion model, or can we just fine-tune a part of it? We compared the UniCon structure shown in Figure 3(d) with a structure that discards part of the diffusion model as shown in Figure 3(e). Figure 4 presents the results of this comparison. The findings indicate that if the complete pre-trained diffusion model is not preserved, there is a significant decline in generative capabilities. This demonstrates that although the diffusion model in UniCon is used only for inference, the generative capabilities it has learned can still be effectively utilized by the UniCon adapter.

## 4.3 COMPARISON

We also compare UniCon with existing adapter designs. For the DiT (Peebles & Xie, 2023) diffusion models, we primarily conduct a direct comparison between UniCon and ControlNet (Zhang et al., 2023). For the SD U-Net diffusion models, we compared UniCon with both ControlNet (Zhang et al., 2023) and T2I-Adapter (Mou et al., 2024). These comparisons of controllable diffusion models were conducted across all datasets.

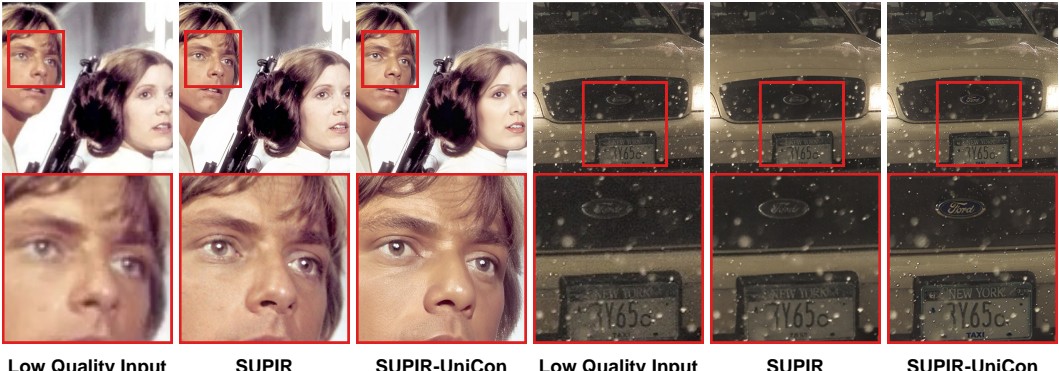

| Low Quality Input | SUPIR | SUPIR-UniCon | Low Quality Input | SUPIR | SUPIR-UniCon |

Figure 8: Results of the SUPIR-UniCon, showcasing UniCon's effectiveness in image restoration.

Table 2 shows the results. It can be seen that our proposed UniCon outperforms the ControlNet and T2I-Adapter in all tasks. In terms of controllability, UniCon excels in maintaining control signals. UniCon also outperforms existing methods in image quality metrics, especially FID scores. It should be noted that although the T2I-Adapter method is better than UniCon in some image quality metrics, the control effect of the T2I method is not good. Combining our ablation studies and comparative experiments, we also found that the scale of the adapter impacts the final outcomes. UniCon-Half, with only half the parameters, performs notably worse than the full-parameter UniCon but still performs better than ControlNet with a comparable parameter number.[1] Notably, even the full-parameter UniCon has a lower training computational cost than ControlNet.

Figure 5 shows some visual comparisons of high-level conditional generation tasks. As one can see, our proposed UniCon excels in generating images with superior detail structure, such as more intricate facial features and fewer artifacts and tears. Additionally, UniCon surpasses other methods in precision control, as demonstrated in the comparison of the second row's first set. Although the sushi rolls are small in scale and complex in structure, UniCon accurately generates sushi rolls at the correct location, faithfully following the control conditions, unlike other methods that fail to strictly adhere to these conditions, and their generative capabilities are inferior to UniCon. Figure 7 further illustrates UniCon's performance in low-level control generation tasks, which demand high fidelity. Our method not only maintains high-quality generation but also strictly adheres to the controls of the input images, effectively generating even the smallest structures. UniCon demonstrates better results in tasks such as image restoration and super-resolution based on diffusion models, showcasing broad application prospects. More results can be found in the Appendix.

UniCon's efficient resource utilization, enhanced control, and high-fidelity performance make it particularly well-suited for low-level vision and image processing tasks. SUPIR (Yu et al., 2024), the first image restoration model based on large-scale diffusion models, currently represents the state of the art in image restoration. However, SUPIR relies on ControlNet, and further scaling under this approach demands immense computational resources, making large-scale expansion impractical. UniCon effectively addresses this limitation. Building on the SUPIR framework, we trained a new SUPIR-UniCon model using SD3 and UniCon. Some of the results, shown in Figure 8, highlight UniCon's potential for broader applications.

## 5 CONCLUSION

This paper presents UniCon, a novel approach tailored for controlling large-scale diffusion models. UniCon leverages unidirectional flow from the diffusion network to the adapter, simplifying the computational process by eliminating the need to compute and store gradients of the diffusion model. UniCon significantly reduces the VRAM requirements and enhances training speeds while maintaining high fidelity in generated images.

---

[1]Note that the UniCon-Encoder design is ineffective. UniCon operation requires information to flow unidirectionally from the diffusion model to the adapter. Duplicating the encoder leaves no adapter to process information in the decoder, compromising generation quality and potentially failing to produce images.

ACKNOWLEDGE

This work was supported by the National Natural Science Foundation of China (Grant No. 62276251) and the Joint Lab of CAS-HK.

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

## A    BROADER IMPACT

Controlled generation technology, as a pivotal innovation in the field of diffusion models, exerts a significant impact across multiple sectors of society. In the creative industries, it enables artists and designers to realize complex visions with unprecedented precision and flexibility, fostering innovation in digital art, design, and multimedia content creation. In commercial applications, controlled generation technology enhances marketing strategies by offering more targeted and dynamic advertising visuals, effectively engaging consumers. Additionally, its influence extends to education and training, where it can revolutionize teaching methods and materials, especially in visually-dependent disciplines, by generating customized educational content and simulations.

Although our system enables artists, designers, and content creators to realize their creative visions through precise control, it is crucial to recognize the potential negative societal impacts that could arise from misuse or abuse, similar to other AI models for image generation and editing. To address these issues, responsible deployment practices, ethical standards, and including special markers in generated images to increase transparency are essential steps in achieving responsible use.

## B    LIMITATIONS

UniCon's training process eliminates the need for the gradient of the base generative model, significantly reducing computational overhead. However, UniCon does not decrease the network's parameter count, resulting in limited speed improvement during sampling compared to DiT-ControlNet.

## C    MORE DETAILS

**Comparison Models.**    In our experiment with the stable diffusion v2.1 (SD), we employ the official versions of the ControlNet (Zhang et al., 2023) and T2IAdapter (Mou et al., 2024) models without any modifications. Since ControlNet is originally designed for U-Net, there is no official version compatible with the DiT transformer-base diffusion model. Therefore, the DiT-ControlNet variant used in our study is a reproduction by PixArt-$\alpha$ (Chen et al., 2023). Following the design principles of ControlNet, the initial halves of the DiT blocks are copied and trained as the controller. The primary difference from the U-Net-based ControlNet is the lack of skip connections in DiT-ControlNet. This makes DiT-ControlNet unable to detach the gradient of the "Encoder" part of the base model, significantly increasing peak VRAM usage.

**Sampling scheme.**    We employ a second-order DPM solver sampler to generate images, with a sampling step of 24. In our PixArt (Chen et al., 2023) experiment, we incorporated null tokens as negative prompts and applied a CFG (Ho & Salimans, 2022) scale of 4.5. Conversely, in the SD v2.1 experiment, the negative prompt included terms such as *"blurring, dirty, messy, worst quality, low quality, frames, watermark, signature, jpeg artifacts, deformed, low-res, over-smooth"*, and we used a CFG scale of 7.5.

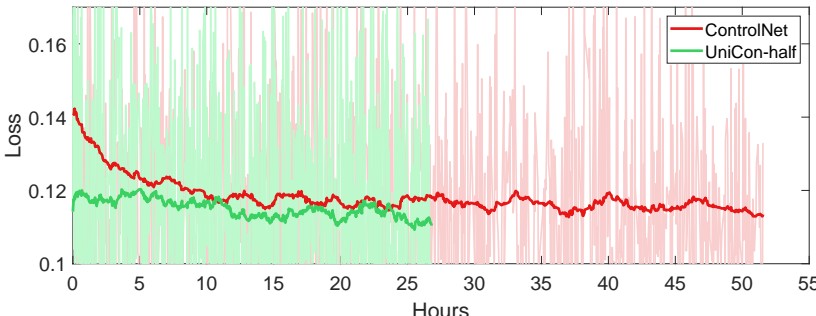

Figure 9: Comparison of training curves between ControlNet and UniCon-half for 100K iterations under Canny edge condition. Notice that ControlNet and UniCon have similar values at the start of training, but show differences after smoothing due to UniCon's faster convergence.

| Task | Training Steps | Controllability | | Generation Quality | | | | Text Consistency |
|---|---|---|---|---|---|---|---|---|
| | | Metric | Value | FID↓ | Clip-IQA↑ | MAN-IQA↑ | MUSIQ↑ | Clip-Score↑ |
| Canny | 50K | SSIM↑ | 0.2974 | 70.40 | 0.4326 | 0.1167 | 50.96 | 0.7623 |
| | 100K | SSIM↑ | 0.5458 | 46.71 | 0.6577 | 0.2029 | 66.45 | 0.7889 |
| | 200K | SSIM↑ | 0.5400 | 46.49 | 0.6529 | 0.2079 | **66.78** | 0.7898 |
| | 400K | SSIM↑ | **0.5507** | **46.13** | **0.6633** | **0.2082** | 66.50 | **0.7933** |
| LQ | 50K | PSNR↑ | 36.48 | 22.86 | 0.6996 | 0.2401 | 68.43 | 0.7909 |
| | 100K | PSNR↑ | 37.34 | 20.34 | 0.7251 | 0.2831 | 69.99 | **0.8022** |
| | 200K | PSNR↑ | **37.40** | 20.07 | 0.7172 | **0.2846** | 69.42 | 0.8014 |
| | 400K | PSNR↑ | 37.35 | **19.94** | **0.7291** | 0.2819 | **69.97** | 0.8009 |

Table 3: Comparisons of different training steps for DiT-UniCon-Full. ↑ indicates the larger the better and ↓ indicates the lower the better. **Bold** represents the best performance.

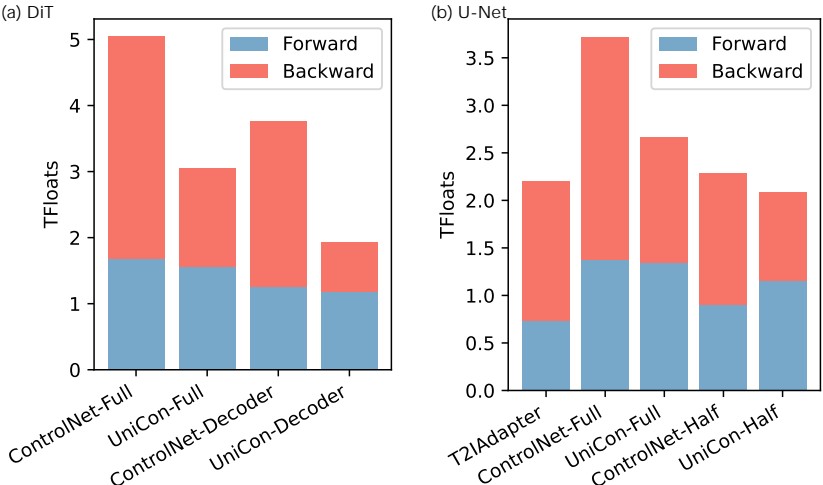

Figure 10: Comparison of (a) DiT and (b) U-Net forward and backward propagating TFloats with different adapters.

**Metric calculations.** We computed PSNR, SSIM, LPIPS, CLIP-IQA, MANIQA, and MUSIQ using the pyiqa library[2], with PSNR specifically calculated on the Y channel. FID scores were determined via the clean-FID[3]. Additionally, MSE was calculated in the RGB domain, with values ranging from 0 to 256. We adopt the mean average precision (mAP) based on object keypoint similarity (OKS) from (Cao et al., 2019; Zhao et al., 2024) as the metric for the pose condition.

**Convergence tendency.** To highlight UniCon's training advantages, we compared its convergence trends with ControlNet. As shown in Figure 9, UniCon retains ControlNet's fast convergence. Despite reversing the direction of information flow, UniCon ensures effective initialization by using a zero-initialized Connector Block and copying parameters from the pre-trained base model.

**Training Steps.** We evaluated the performance of UniCon across varying training steps to identify the optimal step size. As presented in Table 3, UniCon fails to converge at 50K steps for both LQ and Canny tasks. Performance improves substantially from 50K to 100K steps, while further increases beyond 100K yield only marginal gains. Hence, 100K steps is determined to be the most cost-effective training step size.

**Computational evaluation.** As shown in Figure 10, we further analyzed the computational cost of each adaptor during forward and backward propagation. With the same number of learnable parameters, UniCon's total computational cost is lower than ControlNet's. In the UNet-Half setting, UniCon's FP TFloats are slightly higher because it copies the Decoder, which has more parameters

---

[2]https://github.com/chaofengc/IQA-PyTorch
[3]https://github.com/GaParmar/clean-fid

| Task | Adapter | Controllability | | Generation Quality | | | | Text Consistency |
|------|---------|------|------|------|------|------|------|------|
| | | Metric | Value | FID↓ | Clip-IQA↑ | MAN-IQA↑ | MUSIQ↑ | Clip-Score↑ |
| Canny | ControlNet | SSIM↑ | 0.4895 | 49.80 | 0.6683 | 0.2215 | 67.19 | 0.8168 |
| | UniControlNet | SSIM↑ | 0.4996 | 49.95 | 0.6749 | **0.2348** | 67.95 | 0.8152 |
| | GLIGEN | SSIM↑ | 0.4447 | 55.42 | **0.6802** | 0.2280 | 67.91 | 0.7901 |
| | T2I-Adapter | SSIM↑ | 0.3936 | 52.05 | 0.6783 | 0.2266 | **68.16** | 0.8155 |
| | ControlNeXt | SSIM↑ | 0.4568 | 53.03 | 0.6616 | 0.2153 | 66.68 | 0.8141 |
| | UniCon | SSIM↑ | **0.5570** | **47.11** | 0.6704 | 0.2335 | 67.99 | **0.8189** |
| Depth | ControlNet | MSE↓ | 85.70 | 54.30 | 0.6828 | 0.2262 | 67.90 | 0.8202 |
| | UniControlNet | MSE↓ | 85.89 | 54.49 | 0.6846 | 0.2302 | 67.89 | **0.8217** |
| | GLIGEN | MSE↓ | 87.65 | 58.25 | 0.6851 | 0.2234 | **68.41** | 0.7964 |
| | T2I-Adapter | MSE↓ | 87.74 | 55.09 | **0.6906** | **0.2331** | 68.12 | 0.8209 |
| | ControlNeXt | MSE↓ | 86.81 | 55.71 | 0.6799 | 0.2291 | 68.00 | 0.8198 |
| | UniCon | MSE↓ | **85.00** | **53.45** | 0.6807 | 0.2262 | 67.85 | 0.8214 |

Table 4: Comparisons of different adapters for SD-UNet. ↑ indicates the larger the better and ↓ indicates the lower the better. **Bold** represents the best performance.

| Adaptor | Base Model | PSNR↑ | LPIPS↓ | MUSIQ↑ | CLIP-Score↑ | FID↓ |
|---------|-----------|-------|--------|--------|-------------|------|
| StableSR | SD2.1 | 35.39 | 0.0839 | 69.57 | 0.7786 | 37.85 |
| DiffBIR | SD2.1 | 34.81 | 0.0940 | 69.76 | 0.7844 | 32.85 |
| PASD | SD2.1 | 36.17 | 0.0552 | 69.62 | 0.7925 | 25.61 |
| UniCon | SD2.1 | 35.69 | 0.0530 | **70.48** | **0.8027** | 22.80 |
| UniCon | PixArt-$\alpha$ | **37.34** | **0.0453** | 69.99 | 0.8022 | **20.34** |

Table 5: Results of different diffusion-based image restoration models. ↑ indicates the larger the better and ↓ indicates the lower the better. **Bold** represents the best performance.

than ControlNet's copied Encoder. While T2I-Adaptor has the lowest FP TFloats, its control capability is significantly weaker than UniCon's, and its performance degrades noticeably in low-level tasks, as shown in Table 2.

## D  PSEUDO CODES

**Training Cost Experiments.**   In Algorithm 1, we present a pseudo code to evaluate the single-round training cost. The sampling methods for training noise and timestep remain consistent with those used during training. We subsequently report the peak memory usage and average time based on one hundred rounds of training cost evaluation.

**Core Difference between ControlNet and UniCon.**   Algorithm 2 and Algorithm 3 illustrate the pseudo codes for the forward passes of ControlNet and UniCon, respectively. Unlike ControlNet, UniCon avoids storing the base model's gradients during forward propagation and truncates all gradient connections from the base model to the controller. Consequently, the base model gradients are not utilized during backpropagation. These modifications significantly reduce UniCon's memory overhead and training time compared to ControlNet.

## E  MORE COMPARISON RESULTS

**More Adapters on Highlevel Tasks.**   To comprehensively compare UniCon's performance, we evaluated it against other high-level control methods on the UNet model, including UniControl-Net (Zhao et al., 2024), GLIGEN (Li et al., 2023), and ControNeXt (Peng et al., 2024). As shown in Table 4, UniCon achieved the best controllability and FID scores on both canny and depth tasks. NR metrics confirm that UniCon maintains strong control capabilities without notably compromising image quality.

**More Adapters on Lowlevel Tasks.**   Considering that several diffusion-based methods have been applied to low-level tasks, we compared UniCon with related approaches in image restoration (Wang et al., 2023b; Lin et al., 2023; Yang et al., 2023). As shown in Table 5, with SD2.1 as the base

| Adapter | Condition Consistency | | | Image Quality | | | Text Consistency | Diversity |
|---------|-------|-------|--------|-----------|---------|--------|------------------|----------|
|  | PSNR↑ | SSIM↑ | LPIPS↓ | Clip-IQA↑ | MAN-IQA↑ | MUSIQ↑ | Clip-Score↑ | FID↓ |
| ControlNet | 34.82 | 0.9352 | 0.0650 | 0.7147 | 0.2459 | 69.37 | 0.7996 | 26.43 |
| ControlNeXt | 34.31 | 0.9329 | 0.0679 | 0.7171 | 0.2534 | 69.36 | 0.7997 | 26.28 |
| UniCon-Half | 35.64 | **0.9512** | 0.0475 | 0.7042 | 0.2675 | 69.51 | **0.8025** | 22.07 |
| UniCon-Half† | **36.41** | **0.9512** | **0.0462** | **0.7189** | **0.2782** | **69.66** | 0.8008 | **21.34** |

Table 6: Comparisons of different adapters for PixArt-$\alpha$ on ×4 Super-Resolution task. † represents the use of the pre-trained ControlNeXt model as the base model. ↑ indicates the larger the better and ↓ indicates the lower the better. **Bold** represents the best performance.

model, UniCon achieves the highest image quality, with fidelity significantly surpassing StableSR and DiffBIR, and comparable to PASD. Furthermore, using PixArt-$\alpha$ as the base model, UniCon's fidelity improves significantly.

**Discussion of ControlNeXt.**   Since ControlNeXt introduces only a small number of additional parameters during training (Peng et al., 2024), it offers a significant advantage in inference speed compared to other ControlNet variants. However, as shown in Table 6, ControlNeXt's control capabilities are slightly below ControlNet and significantly behind UniCon. While UniCon requires longer inference time, it delivers better performance in scenarios with stricter consistency requirements. Additionally, we found that UniCon and ControlNeXt offer complementary benefits. Using ControlNeXt as UniCon's pre-trained model significantly boosts UniCon's performance, achieving much higher fidelity and quality than ControlNet with the same number of parameters. This two-stage training process can be viewed as a course-to-fine approach with high efficiency.

**Visual Comparison.**   Due to space limitations, we provide more visual comparison results in Figure 15, Figure 16 and Figure 17. Massive visual comparisons prove our method not only maintains high-quality generation but also strictly adheres to the controls of the input images.

# F   FULL ABLATION STUDY RESULTS

Due to space limitations in the main text, we only present some results in the main text. We present the complete ablation experimental results in Table 7, Figure 11, Figure 12, Figure 13 and Figure 14.

**The Effect of Copying "Encoders" and "Decoders" on Diffusion Control.**   Visual comparisons of different adapter structures in Figure 11 reveal that the "Encoders" model exhibits insufficient controllability. In the canny task, this inadequacy manifests as the addition of objects misaligned with the canny lines, such as the giraffe's neck in Figure 11(a). In the SR task, it results in color shifts and artifacts, such as the overall darkness and the sky with messy spots in Figure 11(b). Conversely, while copying "Decoders" improves controllability, it compromises the quality of the generated images, as evidenced by the coloring errors on the face in Figure 11(a) and the messy textures on vegetables and human faces in Figure 11(b). This aligns with the conclusion that focusing on encoders leads to better generation effects, whereas focusing on decoders enhances controllability. As shown in Table 7, the "Skip-Layer" type achieves the highest controllability and image quality among the three copying schemes with the same parameter count. The "Full" copying structure achieves higher metrics and better visual effects by adding parameters on top of the "Skip-Layer" structure, as demonstrated in Figure 11(a), where the "Full" type maintains facial features intact from multiple angles.

**Unidirectional Information Flow vs. Changing Features Directly.**   As indicated in Table 7, using adapters as outputs under the "Skip-Layer" structure negatively impacts the Canny task and causes collapses in the SR task. This likely results from error accumulation in the adapters during initialization with the "Skip-Layer" copying scheme. These errors are less influential when the base diffusion model is used as the output but affect training stability when adapters directly output. Beyond the "Skip-Layer" type, unidirectional information flow steadily improves the model's controllability. In the canny task, unidirectional information flow produces results more consistent with the conditioned lines, as seen in Figure 12(a): the "Full" case 1 curtains and the "Decoder" case 1 water surface. In the SR task, it enhances fidelity, as exemplified by the Chinese string in Figure 12(b). In conclusion, Figure 12 shows that unidirectional information flow significantly reduces training overhead while maintaining overall image quality without degradation.

**The Choice of Connector.**   As demonstrated in Table 7, using ZeroFT as the connector in both the Canny and SR tasks maintains the highest controllability and image quality metrics. Employing ZeroFT enables the network to effectively manage relationships between objects in complex scenes. With ZeroFT, the network can generate a more natural half-length photo of the woman in Figure 13(a) and accurately process the margin between the hollow chandelier and wall in Figure 13(b).

**The Influence of Cross-Attention Blocks in Adapters.**   Previous work, such as BrushNet (Ju et al., 2024), has discussed that removing cross-attention blocks connected to the text prompt in the Adapter module can enhance the fidelity of ControlNet. We investigated whether this approach would also be effective for UniCon. Therefore, we tested the impact of removing cross-attention blocks in the canny and SR tasks in our complete ablation study. As shown in Table 7, removing cross-attention blocks decreases all metrics in the canny task but improves them in the SR task. Visually, dropping cross-attention causes incorrect color matching in the canny task, as seen in Figure 14(a). However, in the SR task, it prevents the generation of cluttered high-frequency textures, as illustrated in Figure 14(b). We speculate that this is because the SR condition already provides sufficient information, making the text prompt less critical, while excessive text guidance can cause CFG to produce messy high-frequency textures. In the canny task, the network relies more on the text prompt to supplement the information that the condition alone cannot provide. Consequently, we removed the cross-attention module in adapters for the SR and Blur+SR tasks.

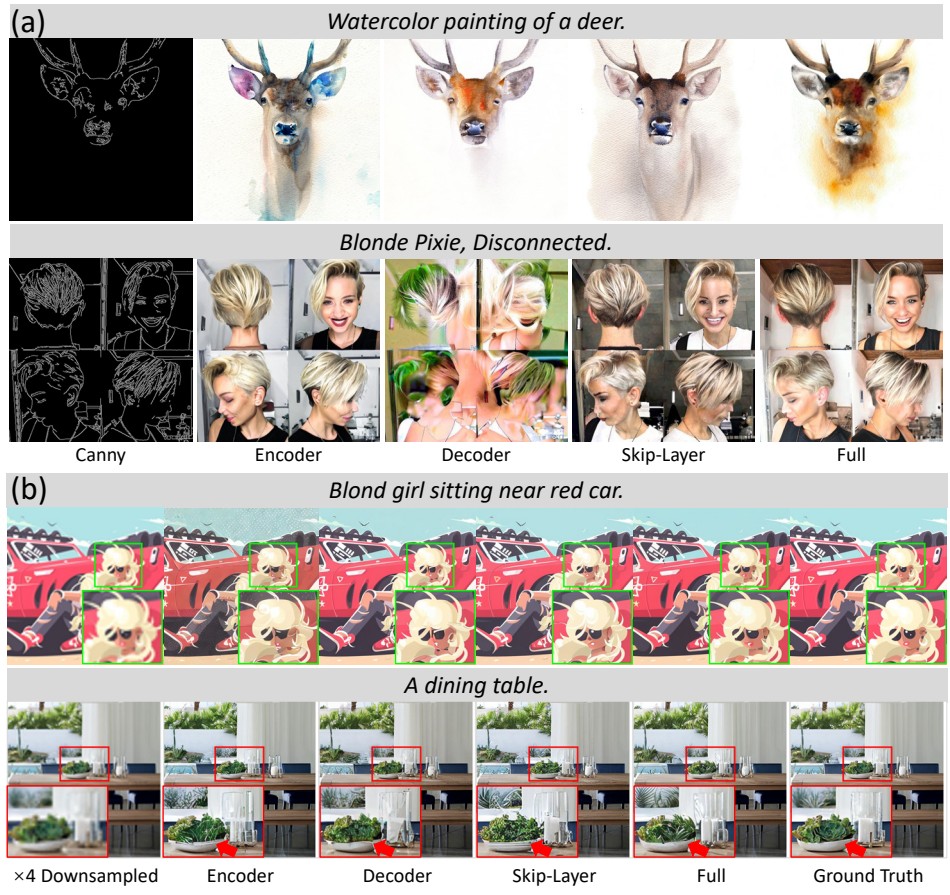

Figure 11: Comparison of different variants of UniCon on conditional generation with (a) Canny and (b) downsampled images.

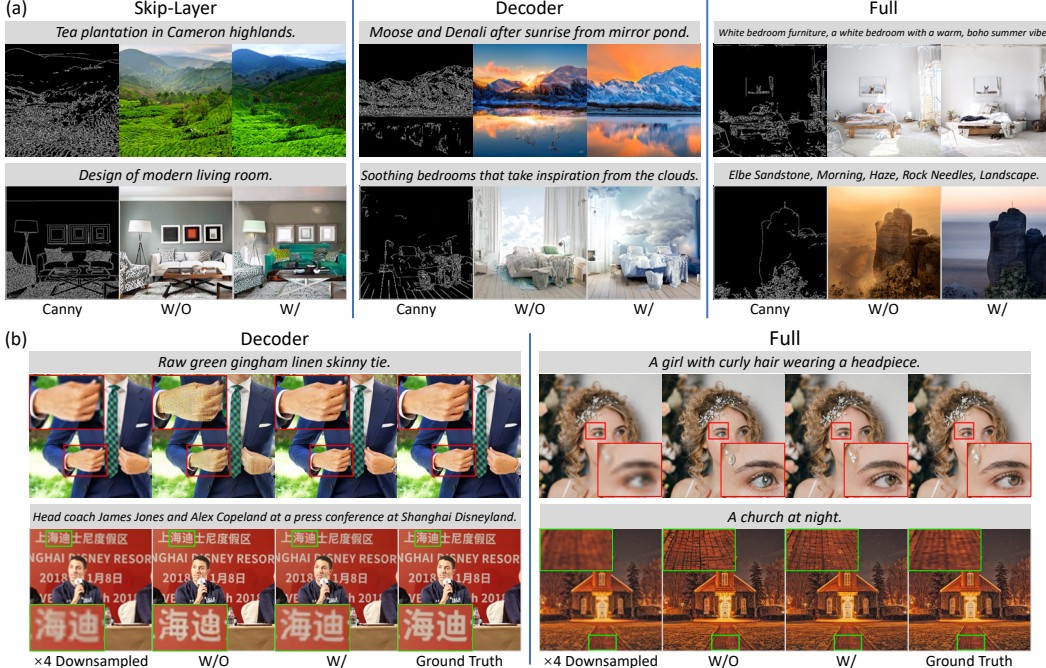

Figure 12: Comparison of different variants of UniCon with or without unidirectional flow on conditional generation with (a) Canny and (b) downsampled images.

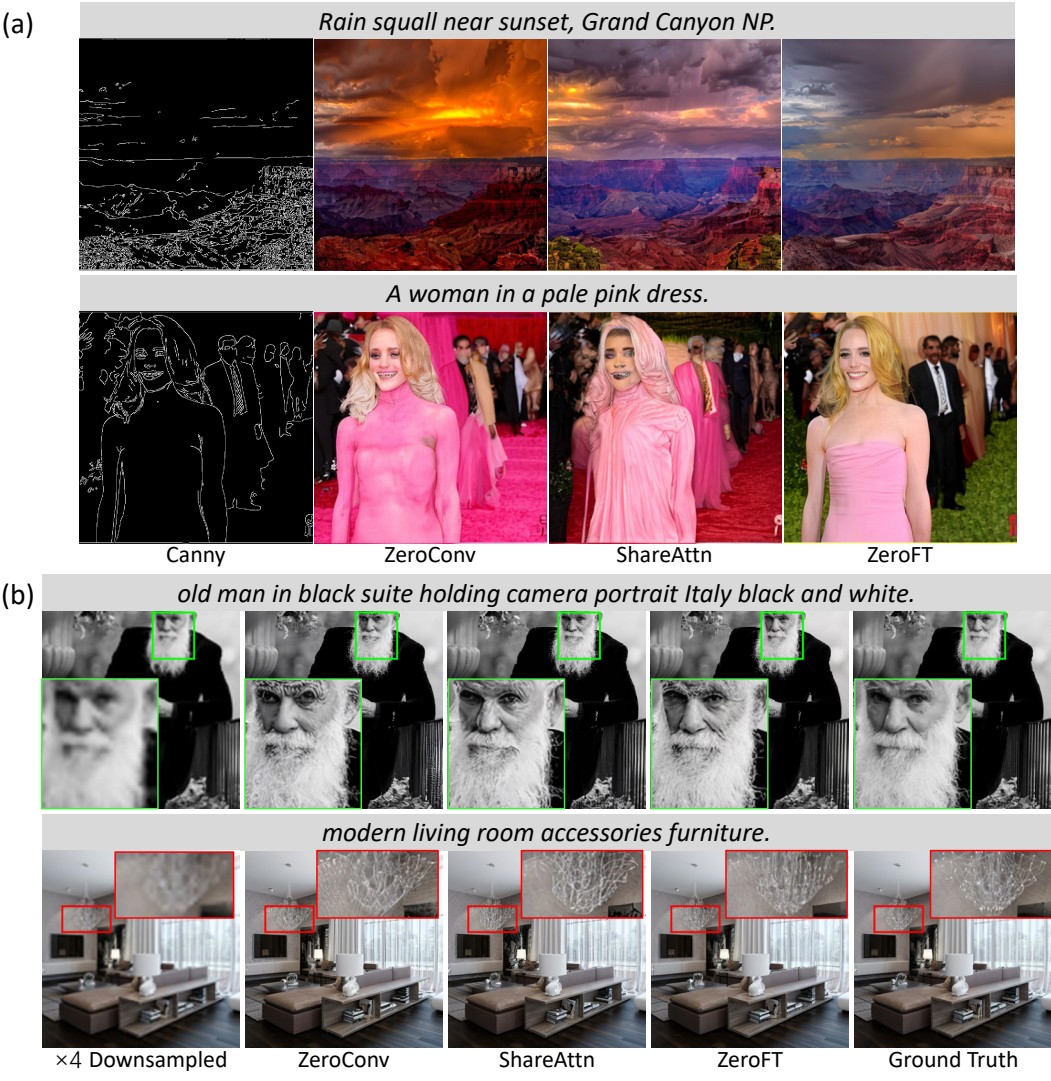

Figure 13: Comparison of UniCon using different connectors on conditional generation with (a) Canny and (b) downsampled images.

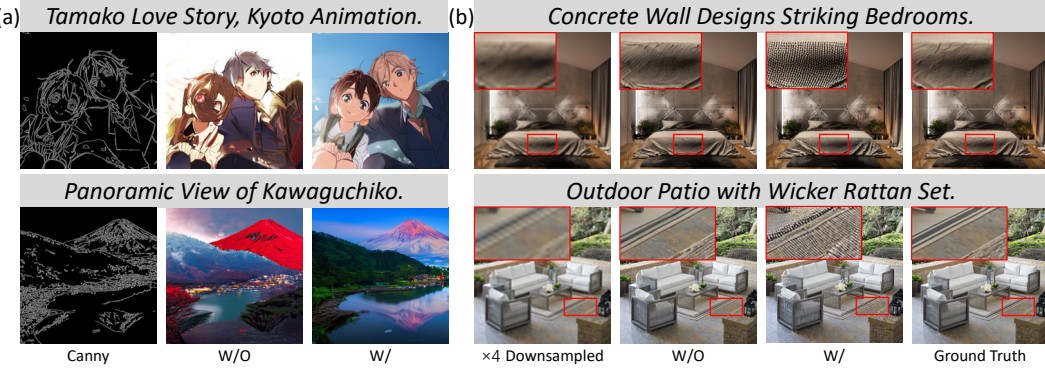

Figure 14: Comparison of UniCon adopting cross attention blocks in Adapter with (a) Canny and (b) downsampled images.

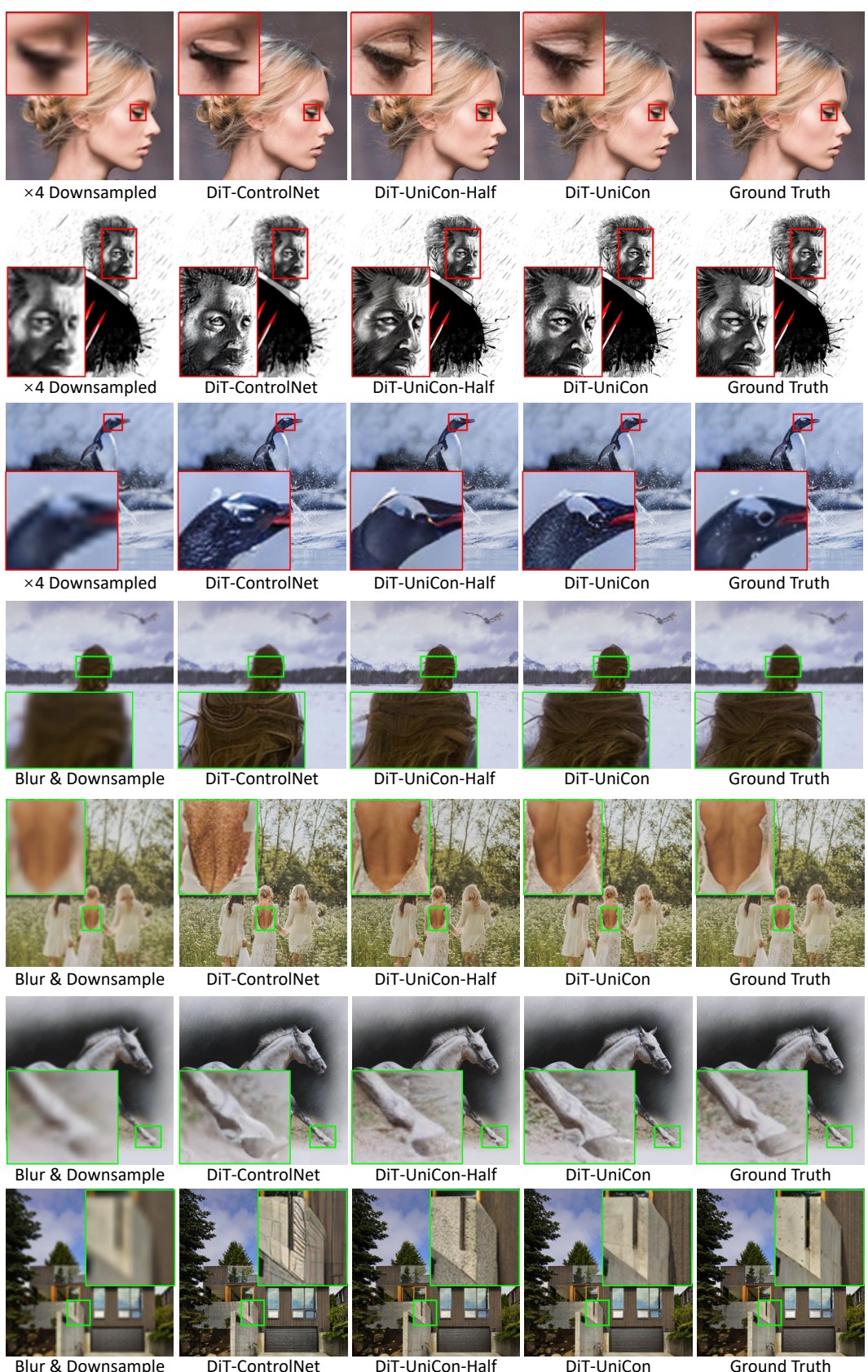

Figure 15: Qualitative comparison of DiT with ControlNet (Zhang et al., 2023) and UniCon dealing with downsampled image condition.

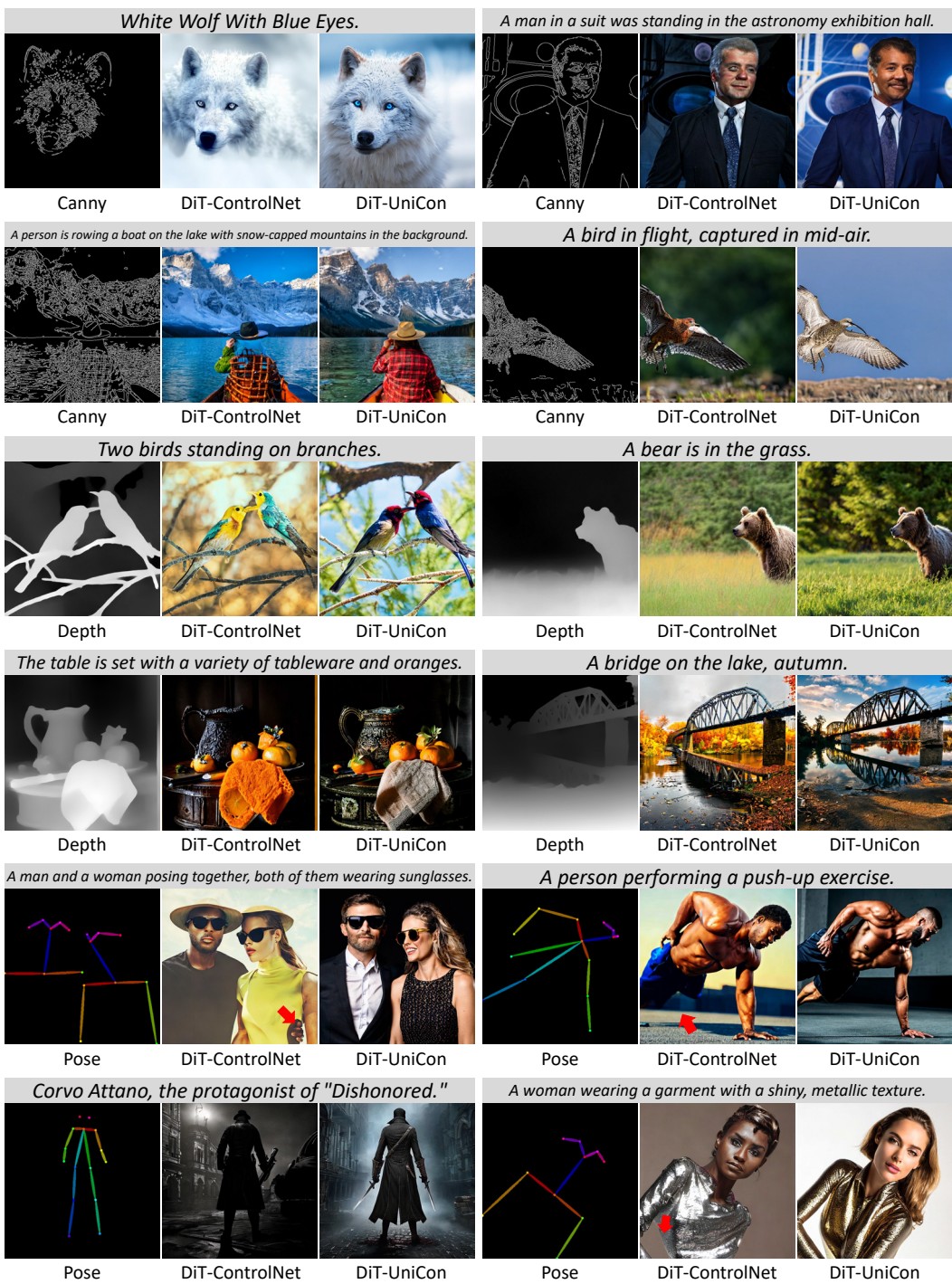

Figure 16: Qualitative comparison of DiT with ControlNet (Zhang et al., 2023) and UniCon dealing with canny, depth, and pose image conditions.

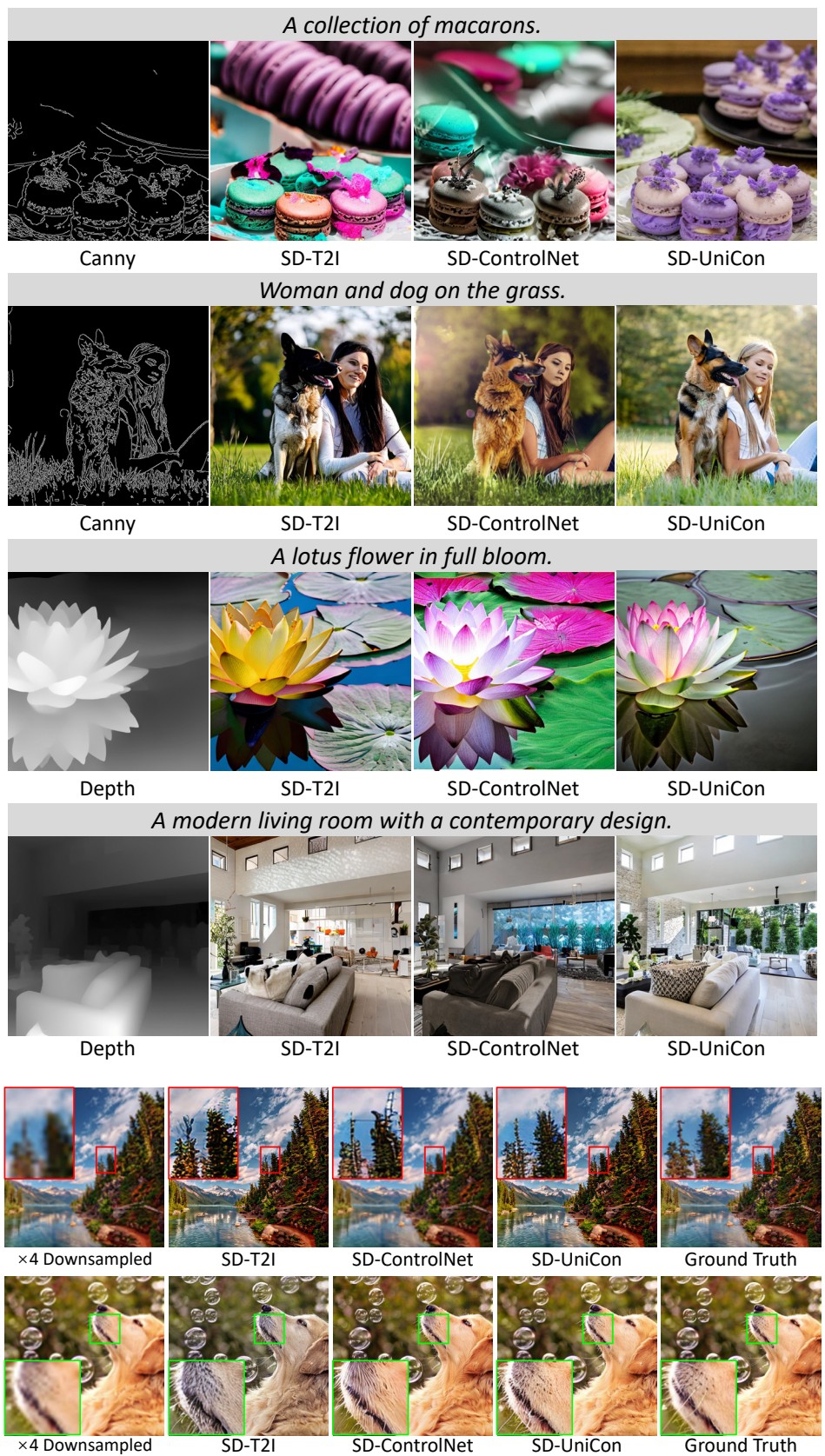

Figure 17: Qualitative comparison of SD with T2I-Adapter (Mou et al., 2024), ControlNet (Zhang et al., 2023) and UniCon dealing with canny, depth, and downsampled image conditions.

| Task | Adapter | Unidirectional | Controller CrossAttn | Connector Type | Controllability PSNR↑ | SSIM↑ | LPIPS↓ | FID↓ | Generation Quality CLIP-IQA↑ | MAN-IQA↑ | MUSIQ↑ | Text Consistency CLIP-Score↑ |
|---|---|---|---|---|---|---|---|---|---|---|---|---|
| Canny | Encoder | ✗ | ✓ | ZeroMLP | - | 0.4748 | 0.3280 | 51.52 | 0.6439 | 0.1861 | 65.24 | 0.7724 |
| | Skip-Layer | ✗ | ✓ | ZeroMLP | - | 0.4983 | 0.3033 | 49.78 | **0.6629** | 0.1978 | 66.05 | 0.7776 |
| | | ✓ | ✓ | ZeroMLP | - | 0.5078 | 0.3016 | 56.93 | 0.6224 | 0.1737 | 63.66 | 0.7561 |
| | Decoder | ✗ | ✓ | ZeroMLP | - | 0.5131 | 0.2953 | 59.32 | 0.6047 | 0.1621 | 62.51 | 0.7507 |
| | | ✓ | ✓ | ZeroMLP | - | 0.5343 | 0.2714 | 55.22 | 0.6347 | 0.1780 | 64.27 | 0.7612 |
| | | ✓ | ✗ | ZeroMLP | - | 0.5260 | 0.2798 | 56.16 | 0.6177 | 0.1638 | 62.40 | 0.7520 |
| | | ✓ | ✓ | ShareAttn | - | 0.5236 | 0.2778 | 56.22 | 0.6154 | 0.1723 | 63.75 | 0.7606 |
| | | ✓ | ✓ | ZeroFT | - | 0.5426 | 0.2658 | 52.31 | 0.6563 | 0.1906 | 66.32 | 0.7696 |
| | Full | ✗ | ✓ | ZeroFT | - | 0.5053 | 0.3070 | 50.17 | 0.6397 | 0.1867 | 64.70 | 0.7818 |
| | | ✓ | ✓ | ZeroFT | - | **0.5458** | **0.2538** | **46.71** | 0.6577 | **0.2029** | **66.45** | **0.7889** |
| DS | Encoder | ✗ | ✓ | ZeroMLP | 34.82 | 0.9352 | 0.0650 | 26.43 | 0.7147 | 0.2459 | 69.37 | 0.7996 |
| | Skip-Layer | ✗ | ✓ | ZeroMLP | 35.49 | 0.9385 | 0.0616 | 24.99 | 0.7229 | 0.2541 | **70.00** | 0.8009 |
| | | ✓ | ✓ | ZeroMLP | | | | Failed | | | | |
| | Decoder | ✗ | ✓ | ZeroMLP | 34.85 | 0.9417 | 0.0589 | 25.84 | 0.6979 | 0.2325 | 68.26 | 0.8013 |
| | | ✓ | ✓ | ZeroMLP | 35.59 | 0.9478 | 0.0519 | 23.55 | 0.7036 | 0.2358 | 68.61 | 0.8018 |
| | | ✓ | ✗ | ZeroMLP | 35.67 | 0.9495 | 0.0502 | 22.99 | 0.7012 | 0.2499 | 69.23 | 0.8013 |
| | | ✓ | ✗ | ShareAttn | 35.55 | 0.9486 | 0.0510 | 23.03 | 0.7078 | 0.2481 | 69.24 | 0.8012 |
| | | ✓ | ✗ | ZeroFT | 35.64 | 0.9512 | 0.0475 | 22.07 | 0.7042 | **0.2675** | 69.51 | 0.8025 |
| | Full | ✗ | ✗ | ZeroFT | 36.53 | 0.9475 | 0.0522 | 23.04 | 0.7212 | 0.2609 | 69.91 | **0.8026** |
| | | ✓ | ✗ | ZeroFT | **37.34** | **0.9542** | **0.0453** | **20.34** | **0.7251** | **0.2831** | 69.99 | 0.8022 |

Table 7: Results of full ablation study. ↑ indicates the larger the better and ↓ indicates the lower the better. **Bold** and underline represent the best and second best performance.

---

**Algorithm 1** Pseudo Code for One Round Training Cost Evaluation

---

**Require:** sampled noise (eta), noised input (xt), timestep (t), prompt (y), condition (c)
 1: device = torch.device(0)
 2: model = model.to(device)
 3: **WEIGHT_MEMORY** = torch.cuda.max_memory_allocated(device)
 4: with torch.no_grad():
        pred_eta = model(xt, t, y, c)
        _ = MSE_Loss(pred_eta, eta)
 5: **ACTIVATION_MEMORY** = torch.cuda.max_memory_allocated(device)
 6: pred_eta = model(xt, t, y, c)
    loss = MSE_Loss(pred_eta, eta)
    loss.backward()
 7: **GRADIENT_MEMORY** = torch.cuda.max_memory_allocated(device)
 8: fp_start_time = time.time()
 9: pred_eta = model(xt, t, y, c)
    loss = MSE_Loss(pred_eta, eta)
10: **FP_TIME** = time.time() - fp_start_time
11: bp_start_time = time.time()
12: optimizer.zero_grad() loss.backward()
    optimizer.step()
13: **BP_TIME** = time.time() - bp_start_time
14: **OPTIMIZER_MEMORY** = torch.cuda.max_memory_allocated(device)
15: **return  *_MEMORY**, ***_TIME**

---

**Algorithm 2** Pseudo Code for DiT-ControlNet Forward Pass

---

**Require:** noised input (xt), timestep (t), prompt embedding (y), condition (c)
 1: x = base_model.image_embedding(xt)
 2: t_emb = base_model.timestep_embedding(t)
 3: c = controller.image_embedding(c)
 4: t_emb_cond = controller.timestep_embedding(t)
 5: x = x + c
 6: **for** base_block, control_block in zip(base_model.blocks, controller.blocks) **do**
 7:     x = base_block(x, t_emb)
 8:     c = control_block(c, t_emb_cond)
 9:     x = x + control_block.connector(c)
10: **end for**
11: pred = base_model.proj_out(x)
12: **return**  pred

---

**Algorithm 3** Pseudo Code for DiT-UniCon Forward Pass

---

**Require:** noised input (xt), timestep (t), prompt embedding (y), condition (c)
 1: **with torch.no_grad():**
        **x = base_model.image_embedding(xt).detach()**
        **t_emb = base_model.timestep_embedding(t).detach()**
 2: c = controller.image_embedding(c)
 3: t_emb_cond = controller.timestep_embedding(t)
 4: **c = c + x**
 5: **for** base_block, control_block in zip(base_model.blocks, controller.blocks) **do**
 6:     **with torch.no_grad():**
            **x = base_block(x, t_emb).detach()**
 7:     c = control_block(c, t_emb_cond)
 8:     **c = c + control_block.connector(x)**
 9: **end for**
10: **pred = controller.proj_out(c)**
11: **return**  pred

---

