# OpenReview forum: "UniCon: Unidirectional Information Flow for Effective Control of Large-Scale Diffusion Models"
_ICLR.cc/2025/Conference — ICLR 2025 Poster_

### Official Review · Reviewer_1fb4 · 2024-11-03

**Soundness:** 2
**Presentation:** 3
**Contribution:** 2
**Rating:** 6
**Confidence:** 3

**Summary:**

This paper presents UniCon, tailored for both UNet-based and DiT-based large-scale diffusion models. The proposed unidirectional flow method is the core method, allowing the adapter alone to generate the final output to reduce VRAM memory consumption. Experiments demonstrate that UniCon achieves good generative performance with lower resource consumption.

**Strengths:**

1. The writing of this paper is good and easy to read.

2. UniCon offers a new approach to integrating adaptors in DiT-based diffusion models.

3. The visualization and quantitative results from the experiments validate the effectiveness of the method.

**Weaknesses:**

1. UniCon designs two different methods for incorporating adaptors in UNet-based and DiT-based diffusion models, as shown in Figure 2. The differences between these methods are significant; for instance, UniCon for UNet adds many concatenation operations and skip connections, while the DiT-based model employs a different serial structure.
   - a) What specific motivations led the authors to design these two distinct structures?
   - b) If the method needs to adapt to other frameworks, such as Mamba-based diffusion or XLSTM-diffusion, would the adaptor structure need to be redesigned? This raises concerns about the generalizability of UniCon.

2. One of the core contributions of this paper is the introduction of the ‘unidirectional’ concept. Unlike ControlNet, which still uses the original diffusion model as output and therefore needs to store diffusion gradients, UniCon modifies this by freezing the original diffusion and using the adaptor as output, thereby reducing VRAM consumption. This makes UniCon fundamentally similar to the LoRA adaptor.
   - a) How does the performance of UniCon compare to LoRA methods?
   - b) Has the UniCon-encoder version been tested? This could be considered a unidirectional ControlNet, and I believe it is an essential comparison to make.

3. I appreciate the extensive ablation studies conducted by the authors on different structures (in Figure 3) to validate the framework's effectiveness; however, the results are not surprising. The full version which has the most parameters, performs the best, which is expected. More importantly, what motivation or significant proof do these ablation studies provide for the design of this method (please do not conclude that the full version is better than other non-full structures)? If none, these results would render the paper more of an empirical study.

4. For both the main experiments and the VRAM comparison experiments, please include comparisons with ControlNet-Unidirectional-Encoder, as this is a core experiment to validate the effectiveness of the unidirectional method.

**Questions:**

Please see the weaknesses.

If I misunderstand any aspects, I welcome the authors to clarify and discuss.

---

> ### Author Response · Authors · 2024-11-16
> **Response to Reviewer 1fb4 (1/2)**
>
> We thank the reviewer for the time and comments. We address the reviewer's concerns point by point in the following response.
>
> `1.a` *"What specific motivations led the authors to design these two distinct structures?"*
>
> `A`:
> We first explain the motivation behind choosing UNet and DiT. UNet and DiT are currently the most mainstream text-to-image model architectures in the AIGC community, possessing a wide user base. Large UNet models reach up to 2.6 billion parameters (such as SDXL), while DiT models can even go up to 10 billion (FLUX). Using ControlNet for conditional generation tuning with these large model architectures incurs excessively high costs. This makes training ControlNet infeasible under many computational conditions. Furthermore, the control performance provided by ControlNet does not meet the requirements. Therefore, we initiated this research.
>
> Our proposed method **is not a specific network architecture but a design philosophy** fundamentally different from that of ControlNet. ControlNet directly applies external influences to the diffusion model through concatenation or zero convolution. In contrast, UniCon’s design philosophy is to extract information only from the diffusion model without feeding it back. This design approach is highly generalizable; any differences in specific network architectures stem entirely from the diffusion model itself. UniCon replicates these structures, simplifying the training process.
>
> It is important to clarify that **ControlNet is also not a single design applicable to various network architectures**. The ControlNet used with UNet involves specific operations that match UNet, while the version used with DiT structures is entirely different and more aligned with DiT’s design. The same applies to UniCon. We have not made specialized designs for DiT and UNet; instead, we have replicated relevant designs based on their respective structures and applied UniCon’s philosophy of unidirectional information flow. In terms of generality, our method is at least on par with the widely used and accepted ControlNet. Moreover, because it saves more training resources, it can be applied to a greater number of diffusion models.
>
> `1.b.` *If the method needs to adapt to other frameworks, such as Mamba-based diffusion or XLSTM-diffusion, would the adaptor structure need to be redesigned?*
>
> `A`:
> When we apply UniCon to other diffusion models, the network structure does indeed change, but this change is not primarily due to our design. We simply replicate the structure of the corresponding diffusion model to serve as the new adaptor. It is worth emphasizing that when ControlNet and LoRA are similarly applied to these new diffusion models, their details also need adjustments. ControlNet likewise needs to copy the corresponding layers of the diffusion model, and the weights modified by LoRA will vary case by case. However, this does not affect the general perception that ControlNet and LoRA are designs with generality. UniCon is a general method that can be applied to various diffusion models.
>
> `2.` *This makes UniCon fundamentally similar to the LoRA adaptor.*
>
> `A`:
> We respectfully disagree with the reviewer’s comments. In our view, there are significant differences among ControlNet, LoRA, and UniCon.
>
> - **ControlNet** computes the influence to be applied to the diffusion model through an adaptor, thereby modifying the intermediate results during the diffusion model’s inference process.
> - **LoRA** directly modifies the weights of the diffusion model; it is essentially a low-rank version of full-parameter fine-tuning of the diffusion model.
> - The core design philosophy of **UniCon** is not to exert any influence on the diffusion model during inference, but to utilize only the inference results of the diffusion model. This is the most fundamental difference among the three.
>
> From the perspective of resources required for fine-tuning, UniCon is also **entirely different** from LoRA. UniCon’s core design does not require computing the gradients of the diffusion model itself. Even if the model is very large, UniCon only needs to compute the gradients of the adaptor, with the diffusion model used solely for inference. However, LoRA must compute the gradients of the diffusion model, and when the diffusion model is extremely large, the training overhead for LoRA remains substantial.
>
> Therefore, we disagree with the reviewer’s view that *“UniCon is fundamentally similar to LoRA.”*

---

> ### Author Response · Authors · 2024-11-16
> **Response to Reviewer 1fb4 (2/2)**
>
> `2.a.` *How does the performance of UniCon compare to LoRA methods?*
>
> `A`:
> Thanks for the question. The LoRA method is **not suitable for conditional control requiring pixel-level correlation**; it is better at holistic control, such as style and character identity. Injecting conditions only into the attention layers (as in LoRA, Addition KV, Direct Tuning) does not enable the network to fully learn the pixel-level correlation between the condition and the ground truth. That is why we did not include LoRA in the comparison at the first time — our method is proposed to provide better control ability other the ControlNet, let alone LoRA.
>
> In contrast, more advanced methods than LoRA, such as ControlNeXt and GLIGEN, not only adjust the parameters of the attention layers but also inject conditions near the input layer through element-wise addition operations. However, as shown in Table 3 of the appendix, UniCon consistently demonstrates comparable image quality and the best controllability under various conditions, outperforming ControlNeXt and GLIGEN. We plan to add the comparison between our method and LoRA and we are very confident the conclusion will be the same.
>
> `2.b.` *Has the UniCon-encoder version been tested? This could be considered a unidirectional ControlNet, and I believe it is an essential comparison to make.*
>
> `A`:
> Thank you for the reviewer’s question, but we need to emphasize that **the design of UniCon-Encoder is fundamentally invalid**. The premise for unidirectional operation to hold is that information must flow unidirectionally from the diffusion model to the adaptor. If we only copy the encoder, then during the flow of information in the diffusion model's decoder, there will be no corresponding adaptor modules to receive the information. This would abandon the generative capabilities of the diffusion decoder, leading to degraded image quality or even the inability to generate images. In this case, the output of the diffusion decoder is unaffected by the condition, leading to a complete lack of controllability. Therefore, **the design of UniCon-Encoder is fundamentally invalid or unreasonable**.
>
> `3.` *What motivation or significant proof do these ablation studies provide for the design of this method.*
>
> `A`:
> We are very pleased that the reviewer pointed out: “The full version which has the most parameters, performs the best, which is expected.” This is the foundation of our work. We aim to introduce more parameters into controlled generation to achieve better results.
>
> However, advanced diffusion models are usually large-scale, with parameters reaching 2B or even 10B. Without our proposed UniCon method, training control generation models with more parameters is infeasible because it exceeds the memory capacity of most GPUs. Even though we know that increasing the number of parameters would yield better results, we couldn’t achieve this without our method.
>
> In our ablation experiments, using the fully copied adaptor UniCon (Full version UniCon) aims to demonstrate that **our method can train models with better performance under the same training overhead**, because the training overhead of the Full version UniCon is the same as that of ControlNet that only copies the encoder.
>
> The UniCon that only copies the decoder (Decoder UniCon) aims to demonstrate that **under the same parameter conditions, our method is still superior to ControlNet**. The parameter count of Decoder UniCon is the same as that of ControlNet that only copies the encoder, but with lower training costs.
>
> If we compare the Full version UniCon with the fully copied ControlNet, their parameter counts are the same, but our results are still better. It should be noted that **the computational overhead of training ControlNet is nearly twice that of our method**.
>
> In summary, **in all cases, we achieve better results with fewer computational resources**.
>
> `4.` *For both the main experiments and the VRAM comparison experiments, please include comparisons with ControlNet-Unidirectional-Encoder.*
>
> `A`:
> Thanks for the reviewer's question. If we could, we would be very happy to include this experiment. However, as discussed in `2.b`, the structure of ControlNet-Unidirectional-Encoder is not reasonable and feasible. We cannot conduct this experiment.

---

> ### Author Response · Authors · 2024-11-19
> **Follow-up discussions with Reviewer 1fb4**
>
> Dear Reviewer 1fb4,
>
> Thank you once again for your valuable time and insightful comments on our manuscript.
>
> We have provided detailed responses to your concerns, which we believe address all the issues you raised. We would greatly appreciate the opportunity to discuss with you whether our responses have satisfactorily resolved your concerns. Please let us know if there are any aspects of our work that remain unclear.
>
> We understand that your time is valuable, and we would be grateful if you could review our responses and share your thoughts at your earliest convenience. Please know that the opportunity for discussion is limited, so your timely feedback is greatly appreciated.
>
> Thank you for your consideration.
>
> Best regards,
>
> Authors

---

> > ### Comment · Reviewer_1fb4 · 2024-11-22
> > **Thank you for your clarification**
> >
> > I carefully read the other reviewers' comments and the authors' rebuttals. I am very grateful to the author for patiently explaining the questions. I think I misunderstood the design of UniCon earlier. The fact is that UniCon eliminates the need to store the gradient of diffusion models during training.
> >
> > I still have one more question. As I mentioned before,
> > >What specific motivations led the authors to design these two distinct structures?
> >
> > The point I really want to ask is, how do you design the specific structure, for example, UniCon for UNet adds many concatenation operations and skip connections, while the DiT-based model employs a different serial structure. What are the detailed motivations for designing those modules? Or are these designs the result of empirical engineering?  **Since the design of UniCon-UNet is so different from the design of UniCon-DiT.** This makes me worry about whether Unicon can generalize enough for future diffusion models, and whether it can be used simply and directly without too much design (e.g., different concat method).
> >
> > After discussing this question, I will raise the score according to the reply. Thanks.

---

> > > ### Author Response · Authors · 2024-11-22
> > > **Further Response to Reviewer 1fb4**
> > >
> > > Thank you for the reviewer’s prompt response! We are pleased that our reply has met your expectations. Below, we provide additional explanations regarding the design details of our framework.
> > >
> > > First, we acknowledge that the complexity in Figure 2b might be due to the way it is drawn. To fit the arrows and modules within the same space, the connections might appear overly intricate. In contrast, Figure 2a is much easier to understand. On the left, we have the blue DiT Block diffusion model, which is duplicated on the right to act as the adaptor. Features from corresponding layers of the diffusion model are injected into the adaptor at matching positions and fused with the adaptor’s features. The fusion method used is ZeroFT. We have added two additional layers: a ZeroMLP at the beginning of the adaptor and a Reprojection layer at the end. The purpose of the ZeroMLP is straightforward—we aim to prevent the conditioning from strongly influencing the adaptor’s training at the start, so we include a zero-initialized layer, similar to ZeroConv in ControlNet. The rationale for the Reprojection layer is even more evident. At the end, the diffusion model’s features are connected to the adaptor, but the ZeroFT method might not be well-suited for direct output. Therefore, we add a learnable transformation layer to handle this connection effectively.
> > >
> > > The structure in Figure 2b might look complex, but it follows the exact same concept as Figure 2a. We first replicate the blue diffusion model on the left to create the backbone of the adaptor on the right. The seemingly intricate connections, shortcut connections, and concatenations are inherent to the UNet design itself, which we have not modified in any way. In the replicated adaptor, we also preserve the original shortcut connections and concatenations. Next, we fuse the features from the diffusion model with the corresponding features in the adaptor. Since the structure is symmetrical, these correspondences are straightforward to establish. The fusion method remains ZeroFT, identical to the design in the DiT structure shown in Figure 2a. Similarly, there is an output layer at the end of the adaptor, but since UNet is a convolutional structure, this output layer is a convolution layer. The only difference is that the UNet adaptor does not include a ZeroMLP at the beginning. This is because UNet does not require a PatchEmbedding operation, making the Zero layer less necessary (although including it would not have any adverse effects).
> > >
> > > In summary, our method is highly general. The core operation involves duplicating the diffusion model and fusing the features of the two networks at corresponding layers using ZeroFT. This simplicity is why we claim that the design can be easily applied to various diffusion models. Some experiments in our paper even demonstrate that it is not necessary to duplicate all layers; duplicating only the latter half, or even duplicating every other layer, still works effectively as long as the corresponding features are fused. This makes our method extremely flexible. As for the additional design elements, they do not alter the fundamental essence of the method; rather, they are refinements added based on our understanding of the network. For instance, omitting the final layers like “Reprojection” or “OutputConv” would not render the method ineffective. UniCon, as a framework, is remarkably robust.
> > >
> > > Thank you for the reviewer’s question. We agree that the clarity of Figure 2 could be improved. We plan to include a clearer version in the supplementary materials, removing the constraints imposed by the limited space in the main text. Additionally, the discussions mentioned above will be updated in the paper.

---

> ### Author Response · Authors · 2024-11-25
> **Further Response to Reviewer 1fb4**
>
> Thank you for your previous feedback. A few days ago, we responded the remaining issue of the architecture design.
>
> As the deadline for the author response is approaching, we would like to take this final opportunity to clarify any potential concerns. If there are additional aspects of our work that require further explanation or elaboration, we would be glad to address them to ensure our contributions are fully understood.
>
> Thank you once again for your thoughtful and constructive comments throughout this process.
>
> Best regards,
>
> Authors

---

> > ### Comment · Reviewer_1fb4 · 2024-11-27
> > **Official Comment by Reviewer 1fb4**
> >
> > Thanks for your reply! I think the authors addressed my main concerns. I hope the authors will explain in detail the generality of UniCon in future editions while minimizing too many complex structural designs to avoid misunderstandings. Therefore, I have raised my score.

---

### Official Review · Reviewer_r7et · 2024-11-04

**Soundness:** 3
**Presentation:** 2
**Contribution:** 3
**Rating:** 8
**Confidence:** 5

**Summary:**

This paper introduces UniCon, a novel approach for effective and efficient control in large-scale diffusion models using unidirectional information flow. The presentation is overall quite good, with clear and informative figures and tables, although there is some room for improvement in the writing accuracy. The experiments appear comprehensive and solid. I look forward to seeing feedback from other reviewers to ensure that I haven’t missed any important aspects.

**Strengths:**

1. The paper is well-structured and easy to follow. Visual representations in figures and tables effectively support the text.

2. The "Unidirectional information flow" concept is novel, ensuring conditional inputs do not interfere with the inherent data structure, a significant advantage.

3. The experiments conducted are comprehensive, validating the proposed method’s efficiency and performance across different tasks and models.

**Weaknesses:**

1. A minor typographical error was noted: "PixeArt" should be corrected to "PixArt" (Line 122).
2. The statement that current designs may not be suitable for transformer-based diffusion models (Line 198-205) could be more thoroughly justified, as it might not hold universally.
3. There is an error in Table 1(a), where an incorrect metric has been highlighted and needs correction.
4. I'm afraid I have to disagree with the conclusion drawn in lines 372-376, as the phenomenon might be attributed to the decoder being a more significant part of the model. In fine-tuned models, there is often a trade-off between image quality and controllability—enhanced controllability may lead to a reduction in image quality.
5. Additionally, "generation effect" does not seem to be the most appropriate term for this context and could be replaced with a more accurate expression.

**Questions:**

* Will the code and weights be released to facilitate broader adoption and further development by the research community?

---

> ### Author Response · Authors · 2024-11-18
> **Response to Reviewer r7et**
>
> We greatly appreciate the reviewer’s recognition of our work, especially the positive comments on our presentation, novelty, experimental design, and results. We next address each of the reviewer’s concerns.
>
> `1.` *“PixeArt” should be corrected to “PixArt”*
>
> `A`: We thank the reviewers for pointing out this typo. We have corrected it and have thoroughly proofread the manuscript to address any related language issues. We will update the paper accordingly.
>
> `2.` *The statement that current designs may not be suitable for transformer-based diffusion models (Line 198-205) could be more thoroughly justified, as it might not hold universally.*
>
> `A`:
> Thanks for the reviewer’s question and sorry for any confusion. Our point is that these control models (ControlNet, UniControlNet, T2I-Adaptor, GLIGEN) **are all optimizations and adjustments based on the UNet design. While applying them directly to DiT might have some effect, it may not yield optimal results.** Currently, there is no work focusing on the micro-design of control in DiT, and therefore no established best practices exist yet. In our work on UniCon, we encountered similar issues; directly applying these designs led to suboptimal outcomes. In our experiments, we introduced some modifications related to the DiT architecture, allowing methods designed for UNet to function stably on DiT. However, **we cannot rule out the possibility that with further design, these methods could achieve better results**, which is beyond the scope of this paper. For instance, as shown in Table 1(a) in the main text, the Skip layer of ControlNet significantly outperforms the Copy Encoder under the original UNet design, even though both methods have the same number of parameters and training overhead. This is due to the lack of an encoder-decoder architecture in DiT. Because DiT does not differentiate between encoder and decoder, we were able to make this modification.
>
> Therefore, directly applying the design of UNet may not necessarily bring the optimal results. Our work is a new attempt on this basis.
>
> `3.` *There is an error in Table 1(a), where an incorrect metric has been highlighted and needs correction.*
>
> `A`: We thank the reviewer for bringing this issue to our attention. We have corrected the typo in the new version and will also review the other tables.
>
> `4.` *the conclusion drawn in lines 372-376. In fine-tuned models, there is often a trade-off between image quality and controllability—enhanced controllability may lead to a reduction in image quality.*
>
> `A`: We agree with reviewer’s comment. Given the same model capacity, there is indeed a trade-off between image quality and controllability. Therefore, the conclusion can be revised as follows: The decoder is the more significant part of the model; since it directly controls the output, controlling the decoder leads to high constraints. In contrast, the conditional constraints imposed by the Copy Encoder are relatively loose, allowing the network to better generate content, which leads to higher generative abilities.
>
> `5.` *“generation effect” does not seem to be the most appropriate term for this context and could be replaced with a more accurate expression.*
>
> `A`: Thank you for your question. We agree that the term “generation effect” is not sufficiently clear. Based on your suggestion, we will rephrase “generation effect” as “image quality.” We believe this change will address your concerns. We are open in this question and willing to discuss further.
>
> `6.` *Will the code and weights be released to facilitate broader adoption and further development by the research community?*
>
> `A`: We thank the reviewer for their suggestion. We will open-source all UniCon model parameters and the training and testing code for all conditions.

---

> > ### Comment · Reviewer_r7et · 2024-11-26
> >
> > Thanks for the detailed response from the authors. It comprehensively addresses all the concerns I raised in my initial review.
> > After carefully reading the reviews from other reviewers, I would like to raise my score.

---

### Official Review · Reviewer_5dEj · 2024-11-04

**Soundness:** 3
**Presentation:** 2
**Contribution:** 3
**Rating:** 6
**Confidence:** 4

**Summary:**

The paper proposed a new method called UniCon, using unidirectional flow for adapting diffusion models. Validation had been made on both the SD U-Net diffusion model and the DiT diffusion model, on several datasets. It turns out that UniCon reached a good result of memory reduction and training speeding up while maintaining the same adapter parameter size and generative capabilities.

**Strengths:**

1. The motivation makes sense: for a faster and more efficient training of controlled generation of diffusion models, towards further scaling up (good for training of scaling up).
2. The evaluation is clear and the experiment results are abundant.
3. Results of unidirectional flow are promising, making a good contribution to the controlled generation of diffusion models and further work.
4. The appendix section of the paper is thorough and contains quality information.

**Weaknesses:**

1. A more detailed introduction could be made of diffusion models for readers' better following. Along with a detailed explanation of ControlNet.
2. The speeding up and memory reduction are mainly for training. But during inference, will UniCon be similar even worse in latency and memory due to the more complex structure (e.g., more Control Blocks compared to the vanilla one on UNet structure, and each block seems to contain more layers of MLP)? This also seems to explain why UniCon performs better because the block structure is more complex.
3. It seems that the page number of Page1, the header and the edge of Page2 can be incorrectly hyperlinked.

**Questions:**

1. Can you provide a schematic or computational rule of forward computation and backpropagation during training to compare ControlNet and UniCon? Just looking at Figure1 and Figure2, readers may not understand why UniCon does not require computing and storing gradients for the diffusion model while ControlNet requires. As in Line 187: "For example, when training ControlNet for DiT, when the gradient of ControlNet itself occupies about 18GB VRAM, the gradient brought by the DiT diffusion model needs to occupy 16GB VRAM."
2. As for the choice of the connector (Line 459) in Table 1b, it seems that each ZeroFT contains $\textbf{2 layers}$ of ZeroMLP (joined by multiplication and addition). Is the ZeroMLP compared here consisting of $\textbf{single-layer}$ structure as the one in the original article? Is it unfair? More rigorous experiment results may be needed to illustrate ZeroFT's effectiveness.

---

> ### Author Response · Authors · 2024-11-19
> **Response to Reviewer 5dEj**
>
> We sincerely thank the reviewer for acknowledging the strengths of our work, including its clear motivation, comprehensive evaluation, promising results, and the quality of the appendix.
>
> `1.` *A more detailed introduction could be made of diffusion models for readers’ better following. Along with a detailed explanation of ControlNet.*
>
> `A`: Thank you for the reviewer’s suggestion. While diffusion models and ControlNet indeed hold significant influence, we will consider adding a detailed description of diffusion models and ControlNet in the supplementary materials. Additionally, we will include references in the related work section that provide more comprehensive information on these topics. We hope this will further assist others in understanding our work.
>
> `2.` *The speeding up and memory reduction are mainly for training. But during inference, will UniCon be similar even worse in latency and memory due to the more complex structure*
>
> `A`: Thank you for your question.
>
> First, even when compared with the same number of parameters, our UniCon design still achieves better results. UniCon-Half has the same parameter count as ControlNet, and their inference speed and memory consumption are similar. However, UniCon-Half has a lower training cost than ControlNet. According to the results in Table 2, UniCon-Half outperforms ControlNet in both control performance and image quality.
>
> Under the same training cost, our UniCon also provides better results. UniCon-Full (UniCon that copies both the Encoder and Decoder) requires the same training resources as ControlNet but has significantly more parameters and achieves better performance. Compared to UniCon-Half, UniCon-Full can be viewed as a scaled-up version that uses more parameters and computation to achieve higher performance. Our proposed UniCon makes this scaling up possible.
>
> **More importantly, without UniCon, training ControlNet for large-scale diffusion models like Flux (10B) or SD3 (2–8B) would be extremely costly**, and most GPUs cannot meet the requirements. UniCon offers a lower-barrier, better-performing solution for controlled generation with such large diffusion models.
>
> `3.` *It seems that the page number of Page1, the header and the edge of Page2 can be incorrectly hyperlinked.*
>
> `A`: We thank the reviewer for pointing out this issue. It was caused by a citation spanning two pages. We have now resolved the problem.
>
> `4.` *Can you provide a schematic or computational rule of forward computation and backpropagation during training to compare ControlNet and UniCon?*
>
> `A`: Thank you for your suggestion. We will consider adding diagrams related to backpropagation in the paper. To provide an intuitive understanding, since ControlNet’s output is applied to the diffusion model, during gradient computation, the gradients need to flow back from the diffusion model to ControlNet. Therefore, we need to compute the gradients of the diffusion model itself. In UniCon, the features from the diffusion model can be regarded as a type of input to the adapter network. When computing gradients, we do not need to calculate gradients with respect to the inputs. Thus, UniCon does not need to store the gradients of the diffusion model.
>
> We have added Algorithms 2 and 3 in the supplementary materials to present the pseudo-code for training UniCon and ControlNet. The key difference is that during the forward pass in UniCon, we do not store gradients for the base model and detach all feature map gradients of the base model.
>
> `5.` *Is comparison of ZeroFT and ZeroMLP unfair?*
>
> `A`: Thanks for the question. Although the three connectors in Table 1b have different numbers of parameters, they represent only a very small portion of the total overhead compared to the base model and controller. For example, in UniCon-Full, ZeroFT and ZeroMLP differ by only 19M parameters, but the entire model has 1.1 billion parameters — a difference of just 1.7%. The primary purpose of our experiments is to assess whether adding a multiplication layer can lead to performance improvements—as observed in SUPIR, where ZeroSFT outperforms ZeroConv.

---

> > ### Comment · Reviewer_5dEj · 2024-12-03
> >
> > Thank you for the response from the authors. I have carefully reviewed the questions raised by other reviewers and the responses provided by the authors, especially the more layman-friendly explanation in the Response to Reviewer Mp3R regarding the potential differences in principles between UniCon and ControlNet. The idea of introducing new structures instead of modifying the Diffusion Models themselves to make the model training and deployment more lightweight is excellent.
> >
> > Overall, I still maintain a stance of acceptance, but my confidence has been improved. I hope that further optimization can be made in the presentation.

---

### Official Review · Reviewer_Mp3R · 2024-11-09

**Soundness:** 4
**Presentation:** 4
**Contribution:** 4
**Rating:** 8
**Confidence:** 3

**Summary:**

The paper presents UniCon, a novel architecture aimed at enhancing control and efficiency in training adapters for large-scale diffusion models, such as the Diffusion Transformer (DiT). Unlike existing methods that involve bidirectional communication between the control adapter and diffusion model, UniCon employs a unidirectional information flow, where information flows only from the diffusion model to the adapter. This enables the adapter to generate the final output independently, reducing computational demands and memory requirements by eliminating the need for the diffusion model to compute and store gradients during adapter training.

**Strengths:**

UniCon’s key strength is demonstrating that gradients through the base diffusion model are unnecessary for effective control, achieving this with a unidirectional flow design. This architecture enables the adapter to independently generate outputs, significantly reducing memory usage and computational costs by omitting gradient calculations for the base model. The work is substantiated with an ablation study for different architectural alternatives.

**Weaknesses:**

I would have liked to see an insightful reasoning, why the unidirectional flow apparently increases the quality of the output. Would the results converge after longer learning times? Is the effect comparable to a "low" rank regularization (somewhat far fetched)?

**Questions:**

The sentence in line 155 "Given a ..." is broken.

---

> ### Author Response · Authors · 2024-11-19
> **Response to Reviewer Mp3R**
>
> We deeply appreciate the reviewer’s recognition of UniCon’s innovative unidirectional flow design and its contributions. We greatly appreciate the reviewer's excellent summary of our work and thank them for their kind evaluation.
>
> `1.` *I would have liked to see an insightful reasoning, why the unidirectional flow apparently increases the quality of the output. Is the effect comparable to a “low” rank regularization (somewhat far fetched)?*
>
> `A`: Thank you for your question. A significant difference between UniCon and ControlNet is that ControlNet achieves control by modifying the features during the inference process of the diffusion model, whereas UniCon generates outputs using another model based on the diffusion model’s inference process. In ControlNet, even after modifying the features during inference, the decoder stage is still processed by the diffusion model that has not been fine-tuned. In contrast, UniCon treats the diffusion model’s inference process as input and, leveraging the knowledge from the pre-trained diffusion model, uses another trainable model to generate the output.
>
> Abstractly speaking, ControlNet fixes the decoder part of the generative model and fine-tunes the inputs to the decoder for the target task, while UniCon fine-tunes a model itself to fit the target task given the input. It is reasonable that a fine-tuned model yields better results.
>
> Furthermore, modifications to the decoder are closer to the output, so introducing control signals at this stage significantly enhances control capabilities. If conditions are only input to the encoder part, the decoder may not strictly adhere to them. This is a major disadvantage for tasks like Canny and SR, which have high control requirements, as reflected in our test results of unsatisfactory control effects.
>
> This is our intuitive understanding of UniCon’s effectiveness. We remain open to further discussion and are willing to engage with the reviewer. We will update the relevant sections of the paper to reflect this discussion if necessary.
>
> `2.` *Would the results converge after longer learning times?*
>
> `A`: Thank you for the question. To address your concerns, we conduct additional experiments. Previously, we reported results after training for 100k iterations in the main text. We have now further trained the model for 500k iterations on the Canny and SR tasks, and the results are presented in Supplementary Table 3. Since UniCon inherits the fast convergence characteristics, the model converges quickly, and 100k iterations of training are sufficient. The gains beyond 100k iterations are marginal. As shown in Figure 8 of the supplementary materials, UniCon converges rapidly, and its training cost is lower than that of ControlNet.
>
> `3.` *The sentence in line 155 “Given a …” is broken.*
>
> `A`: Thank you for pointing out this issue. We will correct it and continue proofreading to ensure accuracy.

---

> > ### Comment · Reviewer_Mp3R · 2024-12-02
> >
> > I would like to thank the authors for their detailed response and the corrections in the paper. I´m more than happy with this contribution, which is expressed in my rating of 8 already.

---

### Meta-Review · Area_Chair_tada · 2024-12-08

**Metareview:**

This paper introduces UniCon, a novel architecture that uses unidirectional information flow for training adapters in large-scale diffusion models, achieving significant computational efficiency and enhanced generative performance. The method reduces VRAM usage, accelerates training, and allows for larger adapters compared to existing approaches like ControlNet.

Strengths:
* Innovative Architecture: The unidirectional flow eliminates the need for gradient computation in the base diffusion model during adapter training, reducing computational costs and GPU memory requirements.
* Scalability: UniCon enables efficient training of adapters with significantly larger parameters than ControlNet, making it suitable for large-scale diffusion models like DiT and SDXL.
* Comprehensive Evaluation: The paper includes ablation studies and comparisons, showing UniCon's advantages in training efficiency, control performance, and generative quality.
* Practical Impact: UniCon provides a scalable solution for training large diffusion models under constrained resources, broadening accessibility for researchers and practitioners.


Weaknesses:
* Presentation: Some reviewers noted areas where the paper's clarity could be improved, particularly regarding the explanation of UniCon's specific structural designs and generalizability to other diffusion architectures.
* Empirical Focus: While experiments are thorough, some ablation studies could benefit from stronger theoretical motivation or justification beyond parameter scaling.
* Generalizability: Concerns about the adaptability of UniCon to non-UNet or non-DiT diffusion architectures were raised, though the authors provided satisfactory clarifications.

Overall, this paper offers a substantial contribution to the field of generative modeling, addressing key challenges in scaling and control for diffusion models. The reviewers unanimously recommend acceptance, citing the paper’s significant contributions to the field of diffusion models. While some areas could benefit from further refinement, the methodology and results are strong enough to merit inclusion in ICLR 2025.

**Additional Comments On Reviewer Discussion:**

Reviewer Comments and Authors' Responses:

Motivation and Generality of Design (Reviewer 1fb4):
* Concern: Distinct structures for UNet and DiT raise questions about generalizability to other architectures.
* Response: Authors clarified that the structural differences reflect replication of the diffusion model’s architecture, not custom designs. UniCon's core design philosophy—unidirectional flow with ZeroFT—remains consistent and generalizable.

Comparison to LoRA and ControlNet (Reviewer 1fb4, 5dEj):
* Concern: Similarity to LoRA and need for experiments comparing to unidirectional ControlNet.
* Response: Authors argued that UniCon fundamentally differs by not modifying or computing gradients for the diffusion model. UniCon-Encoder was deemed invalid due to lack of corresponding decoder modules for generation.

Inference Efficiency and Resource Use (Reviewer 5dEj):
* Concern: Potential inference latency due to UniCon's complexity.
* Response: Authors showed that UniCon maintains comparable efficiency to ControlNet for models with similar parameter sizes but scales better for larger models with improved performance.

Presentation and Clarity (Reviewer r7et, Mp3R):
* Concern: Clarity of explanations and terminology, e.g., "generation effect."
* Response: Authors updated terminology, resolved typographical issues, and improved descriptions of design choices and experimental results.

Theoretical Motivation for Ablations (Reviewer 1fb4):
* Concern: Ablation studies emphasize parameter scaling without deeper theoretical insights.
* Response: Authors clarified that experiments demonstrate UniCon's ability to train larger models with reduced overhead, highlighting practical scalability rather than novel theoretical insights.

---

### Decision · Program_Chairs · 2025-01-22

Accept (Poster)